# Adaptive β-lactam resistance from an inducible efflux pump that is post-translationally regulated by the DjlA co-chaperone

**Jordan Costafrolaz[1], Gaël Panis[1], Bastien Casu[2], Silvia Ardissone[1], Laurence Degeorges[1], Martin Pilhofer[2], Patrick H. Viollier ⓘ[1] ***

1 Department of Microbiology and Molecular Medicine, Faculty of Medicine/Centre Médical Universitaire, University of Geneva, Geneva, Switzerland, 2 Department of Biology, Institute of Molecular Biology & Biophysics, Eidgenössische Technische Hochschule Zürich, Zürich, Switzerland

* patrick.viollier@unige.ch

**Data Availability Statement:** Representative reconstructed tomograms (EMD-18004, EMD-

## Abstract

The acquisition of multidrug resistance (MDR) determinants jeopardizes treatment of bacterial infections with antibiotics. The tripartite efflux pump AcrAB-NodT confers adaptive MDR in the polarized α-proteobacterium *Caulobacter crescentus* via transcriptional induction by first-generation quinolone antibiotics. We discovered that overexpression of AcrAB-NodT by mutation or exogenous inducers confers resistance to cephalosporin and penicillin (β-lactam) antibiotics. Combining 2-step mutagenesis-sequencing (Mut-Seq) and cephalosporin-resistant point mutants, we dissected how TipR uses a common operator of the divergent *tipR* and *acrAB-nodT* promoter for adaptive and/or potentiated AcrAB-NodT-directed efflux. Chemical screening identified diverse compounds that interfere with DNA binding by TipR or induce its dependent proteolytic turnover. We found that long-term induction of AcrAB-NodT deforms the envelope and that homeostatic control by TipR includes co-induction of the DnaJ-like co-chaperone DjlA, boosting pump assembly and/or capacity in anticipation of envelope stress. Thus, the adaptive MDR regulatory circuitry reconciles drug efflux with co-chaperone function for *trans*-envelope assemblies and maintenance.

## Introduction

Bacterial multidrug resistance (MDR) is jeopardizing treatment of bacterial infections with antibiotics [1,2]. The MDR challenge is particularly grave for diderm (gram-negative) pathogens that already possess an intrinsic MDR determinant: a protective outer membrane (OM) that prevents entry of soluble antibiotics [3]. The OM is typically structured as an asymmetric lipid bilayer, with an inner leaflet of phospholipids and an outer leaflet containing lipopolysaccharide (LPS), a charged glycolipid. By contrast, the inner (or cytoplasmic) membrane (IM) is a phospholipid bilayer, but both IM and OM flank the bacterial cell wall (CW; Fig 1A), an important shape-determining structure that is essential for life under most growth conditions.

18005, EMD-18006, EMD-18007, EMD-18008) have been deposited in the Electron Microscopy Data Bank. Raw figures are available on figshare. com under DOI 10.6084/m9.figshare.24032802. Sequence data (Supplemental S1, S3, S4 Data) have been deposited to the Gene Expression Omnibus (GEO) database (GSE225489 accession). Supplemental S2 Data describes the location of the SNPs. The mass spectrometry proteomics data have been deposited to the ProteomeXchange Consortium via the PRIDE partner repository with the dataset identifier PXD040223 and 10.6019/PXD040223. The data from the analysis is deposited in Supplemental S5 Data.

**Funding:** Financial support was obtained from the Swiss National Science Foundation (Schweizerischer Nationalfonds zur Förderung der Wissenschaftlichen Forschung) grant number CRSII5_198737 to PHV. JC was supported by the Fondation Ernst et Lucie Schmidheiny. BC and MP received funding by the NOMIS foundation. The funders had no role in study design, data collection and analysis, decision to publish, or preparation of the manuscript.

**Competing interests:** The authors have declared that no competing interests exist.

**Abbreviations:** AO, acridine orange; AZT, aztreonam; ChIP-Seq, chromatin immunoprecipitation coupled to deep sequencing; CIP, ciprofloxacin; COL, colistin; cryoET, cryo-electron tomography; CV, crystal violet; CW, cell wall; EMSA, electrophoretic mobility assay; EOP, efficiency of plating; EtBr, Ethidium Bromide; HTH, helix-turn-helix; IIR, insertion inverted repeat; IM, inner membrane; IR, inverted repeated; KAN, kanamycin; LC-MS/MS, liquid chromatography followed by tandem mass spectrometry; LPS, lipopolysaccharide; MBL, metallo-β-lactamase; MDR, multidrug resistance; MG, malachite green; MIR, middle inverted repeat; Mut-Seq, mutagenesis-sequencing; NAL, nalidixic acid; NOV, novobiocin; OM, outer membrane; OMP, OM protein; PBS, phosphate-buffered saline; PMF, proton motive force; PVDF, PolyVinyliDenFluoride; PYE, peptone-yeast extract; Rh6, rhodamine 6G; RIR, right inverted repeat; RNAP, RNA polymerase; RND, resistance-nodulation-division; TBDR, TonB-dependent receptor; TBS, Tris-buffered saline; Tn, transposon; WT, wild-type; 2IR, 2 mutations inverted repeat.

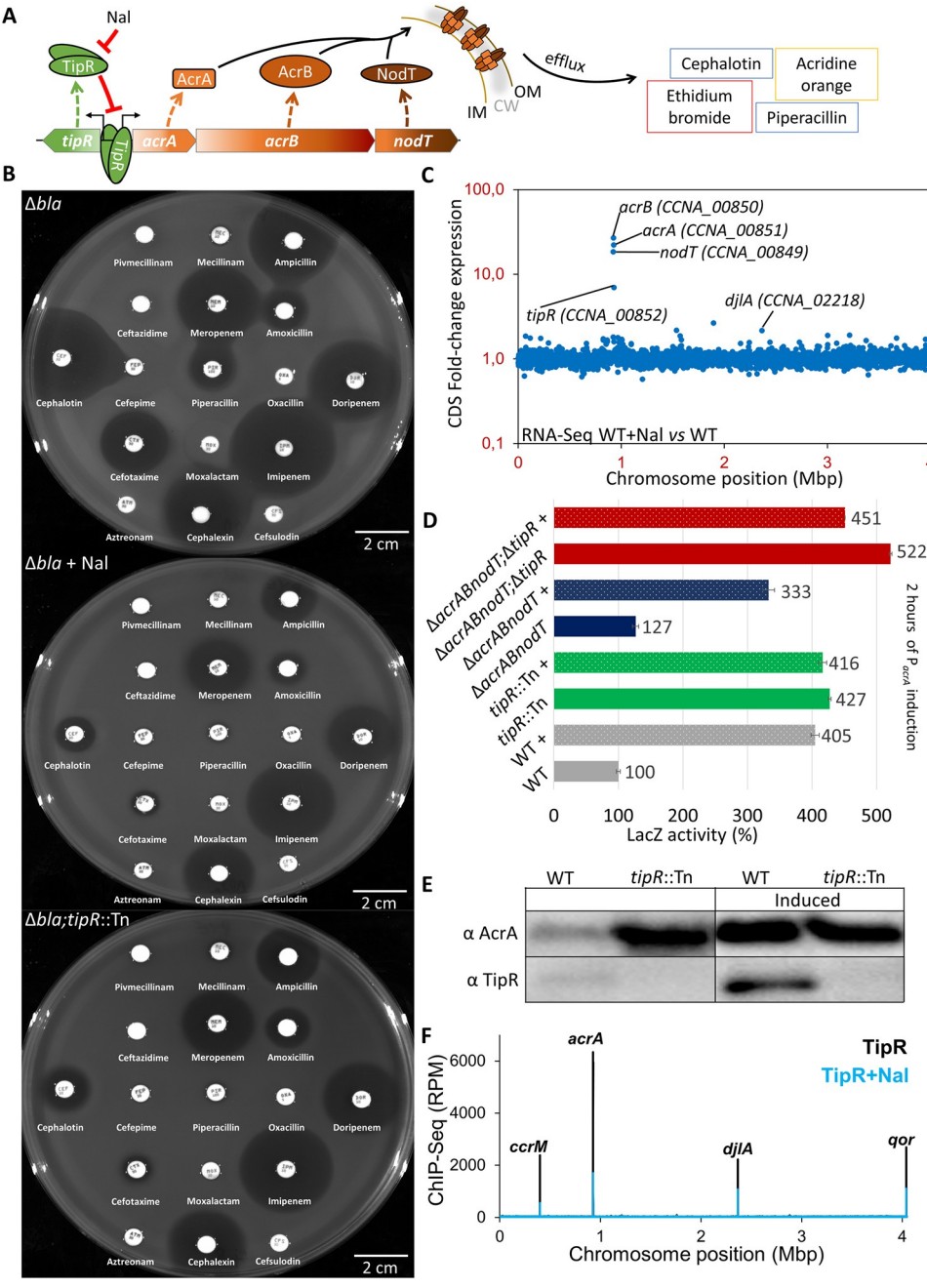

**Fig 1. NAL-dependent derepression of AcrAB-NodT and TipR confers adaptive β-lactam resistance.** (**A**) Schematic of the *tipR/acrA-nodT* locus and the repression of the divergent promoter by TipR, which is antagonized by the quinolone antibiotic Nal. The AcrAB-NodT components assemble into a tripartite complex spanning the IM, CW (grey shading), and OM to expel the diverse compounds indicated on the right via efflux. (**B**) Antibiograms of *Caulobacter crescentus* strains on PYE. Antibiotic discs, from top left to bottom right: Pivmecillinam 20 μg, Mecillinam 10 μg, Ampicillin 100 μg, Ceftazidime 40 μg, Meropenem 10 μg, Amoxicillin 4 μg, Cephalothin 30 μg, Cefepime 30 μg, Piperacillin 100 μg, Oxacillin 5 μg, Doripenem 10 μg, Cefotaxime 30 μg, Moxalactam 30 μg, Imipenem 10 μg, Aztreonam 30 μg, Cephalexin 40 μg, Cefsulodin 30 μg. Nal induction performed at a concentration of 20 μg/mL. (**C**) Dot plot representation of the RNA-Seq analysis realised on *C. crescentus* NA1000 (WT). RNA was extracted from biological replicates of exponential phase cells harvested after induction with and without Nal 20 μg/mL for 30 minutes in PYE. Each dot represents the fold-enrichment of genes expression across the genome (represented as linear starting from the origin of replication) in the Nal-induced condition. The data from the analysis are deposited in S1 Data. (**D**) Activity of β-galactosidase of the P_{acrA}-lacZ in different mutants of *C. crescentus*. The « + » indicates induction by Nal 10 μg/mL for 2 hours. All levels are indicated in percentage of expression regarding the basal level of NA1000 (WT)

without induction. The data from the analysis are deposited in S2 Data. (E) Immunoblot with anti-AcrA and anti-TipR antibodies performed on NA1000 (WT) and *tipR* mutant (*tipR*::Tn) cells with and without induction with of Nal 20 μg/mL for 2 hours. (**F**) Representation of all TipR binding sites on the chromosome of *C. crescentus* (represented as linear starting from the origin of replication). The black line represents the RPM obtained in the condition without induction, while the blue line represent the RPM after 30 minutes of treatment with 20 μg/mL of Nal. Name above indicate the locus. The data from the analysis are deposited in S3 Data. CW, cell wall; IM, inner membrane; Nal, nalidixic acid; OM, outer membrane; PYE, peptone-yeast extract; RPM, reads per million; WT, wild-type.

While the CW is the target of many natural antibiotics, the OM prevents these antibiotics from reaching the CW [4]. Moreover, owing to its unusual lipid composition, the OM is also hydrophilic barrier for hydrophobic antibiotics and as well as large hydrophilic antibiotics that only cross the OM via aqueous pores or nutrient transporters [4,5].

Additional protection against antibiotics that manage to traverse the OM comes from inducible and envelope-spanning multidrug efflux pumps, such as members of the tripartite resistance-nodulation-division (RND) transporter family [5,6]. RND pumps can expel a myriad of different noxious molecules (including antibiotics) from the cell using the proton motive force (PMF). Because RND pump expression can be induced by small molecules, they essentially function confer adaptive MDR. In the uninduced state, transcriptional repressors minimize expression of efflux pumps by binding to their promoters to prevent firing. However, small molecule inducers interact with these repressors, dislodging them from the promoter to relieve repression of the efflux pump gene [7,8] (Fig 1A). Thus, these transcriptional repressors act as single-component chemical sensors for the expulsion of toxic molecules that are encountered and produced, for example, by competing microbes. Although induction of AcrAB-NodT pumps rids the cell from toxic molecules, the massive synthesis and assembly of new *trans*-envelope complexes can also pose (sometimes even lethal) stress to the envelope [9,10]. Thus, protective measures are needed in anticipation of such imbalances, deformations, and/or prolonged stress of the envelope. One possible remedy to avert these problems is to shut off AcrAB-NodT synthesis rapidly after expulsion of the inducer by reinstating promoter repression, but additional mechanisms likely exist.

The best-studied RND-type multidrug efflux pump AcrAB-TolC from the gram-negative γ-proteobacterium *Escherichia coli* has a wide specificity of efflux substrates, including clinically important antimicrobial compounds, dyes, fatty acids, monovalent and bivalent cationic lipophilic antiseptics and disinfectants, detergents, and solvents [5,8,11]. The tripartite system is encoded in the *acrAB-tolC* operon that is repressed by the *cis*-encoded transcriptional repressor AcrR [12]. We recently described the RND-type efflux pump, AcrAB-NodT, which confers adaptive resistance in the oligotrophic α-proteobacterium *Caulobacter crescentus*, a model system for cell cycle and polarity studies. *C. crescentus* AcrAB-NodT is expressed from a promoter ($P_{acrA}$) that is inducible by the first-generation quinolone nalidixic acid (NAL) and related molecules [9,13], and this AcrAB-NodT induction by NAL can be lethal to certain envelope mutants, for example, mutants lacking the TipN polarity factor [9] or the conserved TrcR protein, an RNA polymerase (RNAP)-associated factor [10]. $P_{acrA}$ is repressed by TipR, a TetR-like DNA-binding protein that is divergently encoded and expressed from a shared promoter region that spans 167 nucleotides between the predicted start codons of AcrA and TipR (Fig 1A).

TetR-type transcriptional repressors are characterised by an N-terminal DNA-binding domain with a helix-turn-helix (HTH) motif and a C-terminal regulatory domain interacting with ligands [14]. Binding of an inducer to the C-terminal domain induces a conformational change that ultimately causes its dissociation from the target DNA, liberating the promoter for transcriptional activation by RNAP [15,16]. TetR-like proteins are known to regulate a wide

range of cellular functions, ranging from osmotic and general stress homeostasis, envelope maintenance, virulence gene expression, metabolism and synthesis of antimicrobial compounds, MDR, and expression of efflux pumps [14,17,18]. Whether TipR also has such broad sensory and protective properties has not been explored.

Here, we first report that NAL-based induction of AcrAB-NodT in *C. crescentus* confers cross-protection against penicillins and cephalosporins, β-lactam antibiotics that target CW biosynthesis enzymes. We exploit this relationship in a deep mutational analysis to dissect the molecular determinants and mechanism responsible for TipR-dependent perception of various novel inducers identified in our chemical screens. We also discovered that TipR not only acts in *cis* to control its own expression and that of AcrAB-NodT but additionally targets a distant promoter to repress expression of DjlA, a DnaJ-like co-chaperone that interacts with the DnaK protein folding chaperone [19,20]. Although *C. crescentus* DjlA does not encode a *trans*-membrane anchor as *E. coli* DjlA, we found that long-term AcrAB-NodT overexpression perturbs envelope integrity in *C. crescentus* and that DjlA augments AcrAB-NodT function by interacting with AcrA, thus reconciling the genetically wired co-induction of DjlA and AcrAB via TipR. This circuit design ensures a transient burst of efflux pump and co-chaperone co-expression to prevent efflux pump protein aggregation, while also transiently augmenting efflux capacity in intoxicated cells.

## Results

### NAL-induced adaptive resistance to β-lactams requires AcrAB-NodT

*C. crescentus* Δ*bla* mutant cells lack the chromosomally encoded metallo-β-lactamase (MBL) and are, thus, unable to grow on plates supplemented with certain β-lactams, including the cephalosporin cephalothin (CEF, 10 μg/mL, CEF[10]) or the ureidopenicillin piperacillin (PIR, 40 μg/mL, PIR[40]). To search for determinants conferring β-lactam resistance in cells lacking MBL, we conducted a transposon (Tn) mutagenesis of Δ*bla* mutant cells, delivering the Tn by intergeneric conjugation from an *E. coli* donor. Initially, we counterselected the donor on plates containing NAL (20 μg/mL, NAL[20]), but we unexpectedly observed that Δ*bla* cells grew on PIR[40] or CEF[10] plates supplemented with NAL[20]. NAL is not known to affect β-lactam resistance directly, as it targets the A subunit of DNA gyrase (GyrA) in susceptible bacteria. However, since *C. crescentus* is naturally resistant to NAL owing to the polymorphism F96D in GyrA [9], we speculated that NAL activates a cryptic or an alternative β-lactam resistance mechanism in *C. crescentus*. To confirm that NAL confers β-lactam resistance in Δ*bla* cells, we conducted Kirby–Bauer-based diffusion assays using β-lactam discs placed on plates with or without NAL[20]. Indeed, NAL induces resistance of Δ*bla* cells to various cephalosporins (Fig 1B: CEF, cefotaxime and cephalexin) and penicillins (PIR, ampicillin, and amoxicillin), yet it provided little protection against carbapenems (meropenem, imipenem, and doripenem).

To dissect the genetic basis for the NAL-induced resistance to these β-lactams, we mutagenized Δ*bla* cells with a *himar1* Tn-conferring kanamycin (KAN) resistance, counterselecting the Tn-delivering *E. coli* donor cells with colistin or aztreonam (instead of NAL) because *C. crescentus* is also naturally resistant to these antibiotics. We selected resistant mutants on plates containing KAN[20] and PIR[40] and mapped the Tn insertions of 2 different clones to the *tipR* gene (Fig 2A and 2C) [9,13]. Next, we backcrossed the *tipR*::Tn mutation into Δ*bla* cells and compared the adaptive β-lactam resistance conferred by NAL in Δ*bla*;*tipR*::Tn double-mutant cells to that of Δ*bla* parental cells by Kirby–Bauer antibiotic disc diffusion assays (Fig 1B), as well as by growth on plates containing CEF[10] or PIR[40] (Fig 2C). These tests revealed an identical β-lactam resistance spectrum between the Δ*bla* strain grown in the presence of NAL and

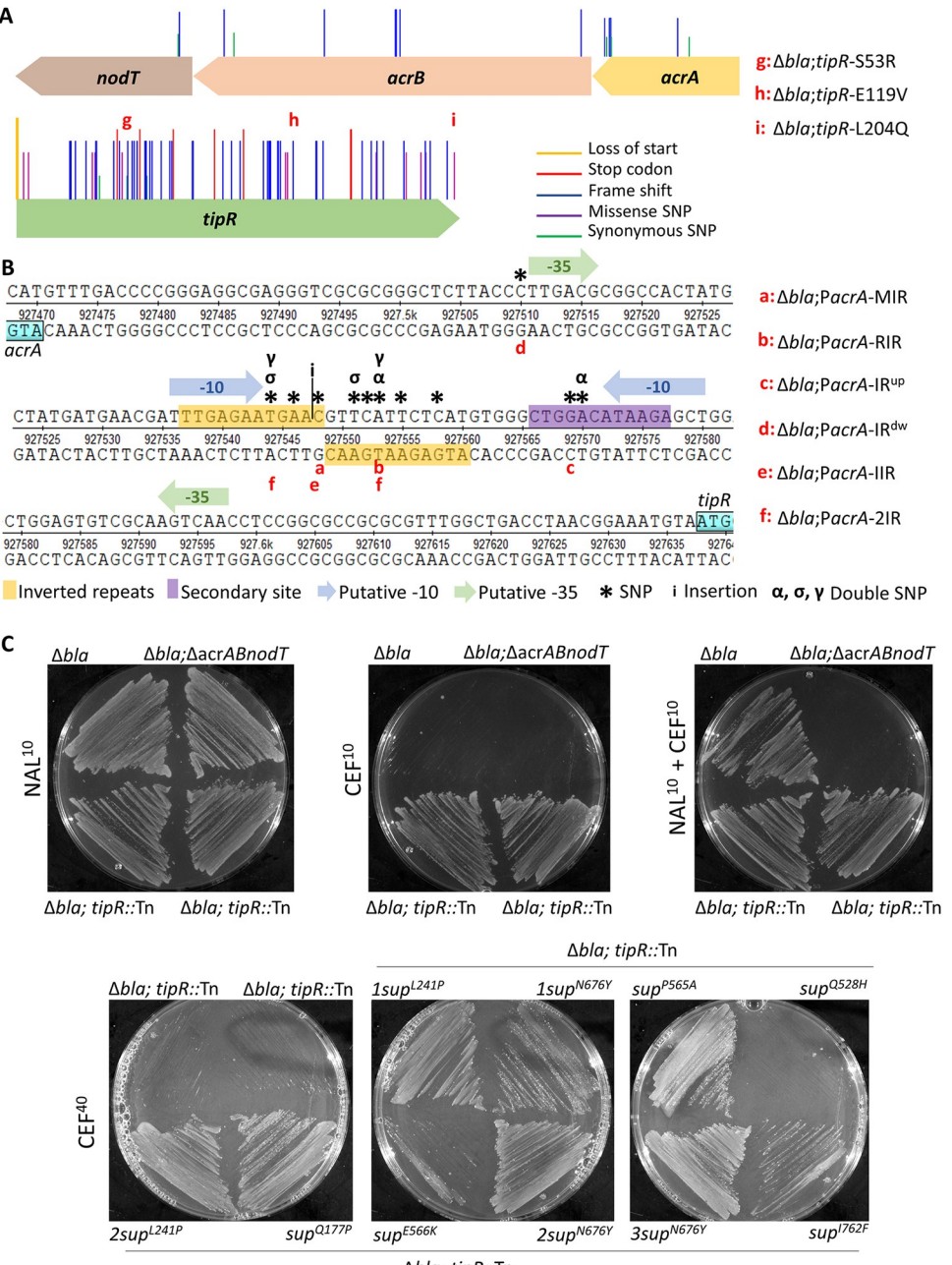

**Fig 2. Deep mutational dissection of the *tipR-acrAB-nodT* locus.** (**A**) Positions of mutation in *tipR* and the *acrAB-nodT* operon conferring resistance to cephalothin (10 μg/mL) in Δ*bla* cells. The list of mutations is a combination of sequences from the Mut-Seq experiment and single-clone sequencing. The line represents, from the tallest to the smallest, the loss of start codon, new stop codon, missense mutations, frameshift, and a synonymous variant. (**B**) Intergenic region between *acrA* and *tipR*. The inverted repeats region is a hot spot for mutations that confer resistance to cephalothin (10 μg/mL) to Δ*bla* cells. The * represent SNPs; (α, σ, γ) are double mutations per strain and (i) insertion. Position of the putative −10 and −35 sequences was predicted using SoftBerry. (**C**) Resistance of Δ*bla* cells and various mutant derivatives to cephalothin or piperacillin. Strains were straked plates containing 10 μg/mL or 40 μg/mL of cephalothin (CEF[10] or CEF[40]), or 40 μg/mL of piperacllin (PIR[40]) with or without NAL (10 μg/mL, NAL[10]) and grown for 72 hours at 30˚C. Note that Δ*ABT* is used as abbreviation for the Δ*acrAB-nodT* mutation.

the Δ*bla*;*tipR*::Tn strain without NAL, indicating that NAL responsiveness is lost without TipR (Figs 1B and 2C), the transcriptional repressor of P$_{acrA}$.

RNA-Seq analysis of NAL-induced cells revealed a 20-fold induction of *acrAB-nodT* transcripts compared to the (uninduced) control state (Figs 1CB and S2A and S1 Data) and a 7-fold induction of *tipR* transcripts, but also a 2-fold induction of transcripts encoding the DnaJ-like co-chaperone DjlA (*CCNA_02218*; see below). Moreover, several transcripts encoding components of respiratory chain/cytochrome biosynthesis pathway (CCNA_01767, CCNA_2850, CCNA_3065) were up-regulated between 1.9- and 1.6-fold. Thus, the transcriptional response to NAL predominantly affects the *acrAB-nodT* and *tipR* locus. No signature of a general transcriptional response to altered DNA supercoiling or DNA damage was apparent, a response that typically occurs in the presence of certain quinolone antibiotics. We hypothesized that the induction of the *djlA* transcript could reflect a direct interplay with TipR and AcrAB-NodT (see below), whereas the weak up-regulation of transcripts encoding respiratory proteins likely stems from increased respiratory demand owing to PMF consumption by elevated AcrAB-NodT-mediated efflux arising from TipR derepression.

To confirm that NAL interferes with TipR-mediated repression of P$_{acrA}$, we conducted β-galactosidase (LacZ)-based promoter probe assays (Fig 1D and S2 Data) in cells transformed with a P$_{acrA}$-*lacZ* transcriptional promoter probe plasmid (pP$_{acrA}$-*lacZ*). In wild-type (*WT*) cells, the addition of NAL lead to a 4-fold induction of P$_{acrA}$-*lacZ* activity (405%) relative to the reference state without NAL (set to 100%). By contrast, P$_{acrA}$-*lacZ* activity was 427% in *tipR*:: Tn cells grown without NAL and remained unchanged upon the addition of NAL. Matching results were obtained by monitoring the accumulation of AcrA by immunoblotting using polyclonal antibodies to AcrA (Figs 1E and S3A). AcrA abundance was strongly induced upon the addition of NAL in *WT* cells. By contrast, AcrA steady-state levels are elevated in *tipR*::Tn cells and not further augmented by the addition of NAL. We conclude that NAL induction of P$_{acrA}$ is likely governed through inactivation or removal of TipR. Moreover, and consistent with the induction of *tipR* mRNA by RNA-Seq, immunoblotting using polyclonal antibodies to TipR revealed a congruent accumulation of TipR and AcrA after 20 minutes during a time course analysis with NAL[10] (S3B Fig), suggesting that *acrA* and *tipR* promoters are simultaneously derepressed upon the addition of NAL.

Next, we conducted ChIP-Seq experiments using polyclonal antibodies to TipR in pursuit of TipRs promoter occupancy in vivo. We discovered that TipR significantly binds 4 chromosomal targets in vivo. Importantly, TipR occupancy of these sites is reduced by ≈500% after exposing cells to NAL (for 30 minutes; Figs 1F and S2A and S3 Data), indicating that NAL antagonizes repression of P$_{acrA}$ and P$_{tipR}$ by TipR in vivo, allowing recruitment of RNAP (and TrcR [10]). The coordinated induction of AcrAB-NodT and TipR by NAL points towards a homeostatic control mechanism in which the adaptive and transient resistance to β-lactam antibiotics is conferred by induction of AcrAB-NodT, followed by TipR synthesis to reestablish the repressed ground state once the inducers have been expelled by AcrAB-NodT.

In support of this notion, we found that NAL no longer confers adaptive β-lactam resistance in Δ*bla* Δ*acrAB-nodT* cells (Figs 2C and S1). While this result indicates that the adaptive resistance is mediated by NAL-based induction of AcrAB-NodT, the basal level expression of AcrAB-NodT in the uninduced state suffices to confer resistance to the fourth-generation cephalosporin cefepime. Such residual constitutive resistance by AcrAB-NodT is also detectable for low levels of cephalothin (first generation) and cefotaxime (third generation; S1B Fig). In the case of the former, we asked if AcrAB-NodT could even confer resistance to higher levels of cephalothin than the resistance arising from inactivation of *tipR* (CEF[10]). To this end, we conducted a second step selection for CEF[40] (cephalothin 40 µg/mL) resistance of Δ*bla*;*tipR*:: Tn cells and isolated several such mutants, each harbouring single missense mutations in *acrB*

(Q177P, L241P, Q528H, P565A, E566K, N676Y, I762F; Figs 2C and S2B). We surmise that these mutations likely increase the specificity, capacity, or abundance of AcrAB-NodT to expel cephalothin (Fig 2C) and that they specifically point to AcrB as a key plasticity determinant enabling high-level MDR and the main element underlying NAL-dependent adaptive resistance (to cephalothin) in *C. crescentus*.

## Ectopic long-term AcrAB-NodT induction perturbs envelope structure and function

Having dissected the role of AcrAB-NodT in adaptive β-lactam resistance, we probed for elevated resistance conferred by the *tipR*::Tn mutation compared to the parent by Kirby–Bauer-based antibiotic disc diffusion assays using a diverse panel of discs with different classes of antibiotics. We observed increased resistance toward the quinolones sparfloxacin and ciprofloxacin as well as to the macrolide antibiotics erythromycin and azithromycin, likely owing to increased efflux conferred by AcrAB-NodT overexpression (S3 Fig). Surprisingly, these experiments also revealed that *tipR*::Tn cells are more susceptible CW-targeting antibiotics such as vancomycin, teicoplanin, bacitracin, and fosfomycin (S4 Fig). As the first 3 molecules are large soluble antibiotics that are usually poorly active against gram-negative bacteria due to the OM, which prevents their entry into cells [21], we wondered whether envelope integrity might be compromised in *tipR*::Tn cells, facilitating the entry of these antibiotics. Indeed, phase contrast microscopy shows frequent envelope blebs and other cell deformations on *tipR*::Tn cells. While such blebs were seen less frequently when AcrAB-NodT induction occurred through NAL than through inactivation of TipR or through simple AcrAB-NodT overexpression form a plasmid (S4A Fig, red arrows), we suspect that NAL-dependent induction could be a transient phenomenon and/or the induction is attenuated because of the expulsion of NAL by AcrAB-NodT. In support of this idea, we found that the envelope deformations do not require the efflux activity of the pump, as cells exposed to the efflux pump inhibitor 1-(1-Naphthyl-methyl)-piperazine (NMP) still harbour these deformations (S5A Fig). Thus, a strong induction and subsequent assembly of this *trans*-envelope structure is sufficient to disfigure cells.

To investigate the nature of those perturbations in detail, we imaged cells by cryo-electron tomography (cryoET). We imaged Δ*bla* cells overexpressing AcrAB-NodT either via *tipR*::Tn mutation and via a multicopy plasmid carrying *acrAB-nodT* (pSRK-*acrAB-nodT*) coupled with native induction of AcrAB-NodT from the chromosomal locus by NAL[10] for maximal overexpression. The cryo-electron tomograms of Δ*bla* cells showed a canonical cell envelope architecture, with a continuous IM, OM, and a paracrystalline surface (S-layer) layer (S5B Fig). In contrast, Δ*bla;tipR*::Tn and Δ*bla;*pSRK-*acrAB-nodT* cells revealed abnormal cell shapes, manifesting in bulging of the entire cell envelope (S6A Fig). Furthermore, these mutants also showed significant perturbations of the structure of the cell envelope, including membrane blebs and diverse types of vesiculations of the membranes (S5B and S6 Figs). While the precise content and origin of those vesicles remains unknown, we suspect that they contain phospholipid- and/or LPS-derived OM material and/or soluble periplasmic content.

In summary, these structural defects are consistent with increased susceptibility toward antibiotics that are typically kept at bay by the OM, and they highlight the necessity for tight and transiently regulated control by TipR for an efflux system that should be induced only under life threatening conditions.

## Genetic dissection of TipR and its target

To dissect the molecular mechanism of induction by TipR, we took advantage of the chemical-genetic relationship between AcrAB-NodT induction and CEF[10] resistance. With the goal of

discovering loss-of-function mutations in *tipR*, or potentially its (unknown) target sequence in $P_{acrA}$, we isolated spontaneous $CEF^{10}$-resistant point mutants by plating Δ*bla* cells on $CEF^{10}$ plates. After collecting approximately 10,000 $CEF^{10}$-resistant Δ*bla* colonies, we PCR amplified *tipR* and *acrAB-nodT* from the mutant pool and deep sequenced the PCR products. Sequence analysis (Fig 2A and S4 Data) of this Mut-Seq dataset indeed revealed missense, nonsense, and frameshift mutations scattered throughout *tipR*, with a slight hotspot in the region predicted to encode the N-terminal DNA-binding domain. The fact that nonsense and frameshift mutations are also found in the C-terminal half of the protein indicates that these residues are also important for TipR function. Interestingly, we also found 14 (gain-of-function) mutations within the *acrAB-nodT* coding sequence that presumably increase the stability, structure, and/ or affinity of AcrAB-NodT components towards CEF (matching the $CEF^{40}$ experiments described above).

Next, we isolated individual $CEF^{10}$-resistant Δ*bla* mutant clones from this pool and sequenced their genomes. Three TipR missense mutants were chosen for further study, each encoding a different mutation in TipR:S53R (in the DNA-binding domain), as well as E119V and L204Q, respectively in the central region and C-terminal domain (Fig 2A, corresponding to mutants Δ*bla;tipR-S53R*, Δ*bla;tipR-E119V*, and Δ*bla;tipR-S53R*). We confirmed the strong increase in $P_{acrA}$ activity in all these mutants using pP$_{acrA}$-*lacZ*-based assays (S7 Fig and S2 Data).

In addition to these loss-of-function mutations in TipR, we also obtained clones with a mutation in the operator site for the promoter of *tipR* (P$_{tipR}$, since *tipR* and *acrAB-nodT* are transcribed divergently from the same intergenic region; Fig 2B) that would prevent TipR from binding and repressing $P_{acrA}$. Since the TipR-binding site sequence is shorter than the 615-nucleotide *tipR* gene, mutations in the former would likely surface less frequently than nonsense mutations in *tipR*. Nonetheless, genome sequencing of $CEF^{10}$-resistant mutants revealed 2 strains with mutations in $P_{acrA}$ (strains Δ*bla* P$_{acrA}$-*RIR* and Δ*bla* P$_{acrA}$-$IR^{up}$), attesting to the strength and specificity of the selection. To favour the enrichment of additional loss-of-function mutations in the TipR-binding site (Fig 2B), we modified our starting genotype for a revised screen. To this end, we conducted the $CEF^{10}$-resistant selection with Δ*bla* cells carrying a multicopy plasmid expressing TipR (pSRK-Gm-*tipR*) to eliminate mutants with a dysfunctional *tipR* gene that would otherwise surface. With this strategy, we uncovered strains with additional $P_{acrA}$ mutations, 80% of which reside in a region of dyad symmetry comprising an 11-base pair inverted repeated (IR, 5′-TGAGAATGAAC-3′; Fig 2B), 2 adjacent mutations are 11 nucleotides upstream of the IR ($IR^{up}$ mutation), and another mutation lies 33 nucleotides downstream of the IR (mutation $IR^{dw}$).

Lastly, we chose the following IR mutant strains for follow-up confirmation studies: Δ*bla* P$_{acrA}$-MIR (middle inverted repeat), Δ*bla* P$_{acrA}$-RIR (right inverted repeat), Δ*bla* P$_{acrA}$-2IR (2 mutations inverted repeat) and Δ*bla* P$_{acrA}$-IIR (insertion inverted repeat; Fig 2B), as well as one with a mutation upstream of the IR (Δ*bla* P$_{acrA}$-$IR^{up}$) and one downstream of it (Δ*bla* P$_{acrA}$-$IR^{dw}$; Fig 2B). Immunoblotting using polyclonal antibodies to AcrA and TipR confirmed that AcrA is overexpressed in all selected mutants compared to the Δ*bla* parent (S8A–S8D Fig), explaining why these mutants grew on $CEF^{10}$ plates. TipR was also overproduced in all mutants to similar levels, except for strains Δ*bla* P$_{acrA}$-$IR^{up}$ and Δ*bla* *tipR-L204Q* (S8A and S8B Fig) in which the TipR levels resemble the basal amount of the Δ*bla* parental strain. The addition of NAL led to the induction of TipR in these 2 mutants, albeit to lower steady-state levels compared to Δ*bla* cells, whereas AcrA levels were not substantially affected. This result suggests that the effect of the mutation differs for $P_{acrA}$ versus P$_{tipR}$ and/or that NAL affects the abundance of TipR through a post-translational mechanism. Below, we provide additional evidence for the latter model.

In summary, the strong selection for CEF[10] resistance in Δ*bla* cells unveiled mutational hotspots in the region predicted to be required for DNA binding by TipR, but also in the IR of in the suspected promoter target of TipR.

## Defining the TipR target sequence and functional residues in TipR

We used electrophoretic mobility assays (EMSAs) to confirm that TipR binds the IR in vitro. In these experiments, a Cy5-labelled P$_{acrA}$ probe centred on the IR was mixed with recombinant (untagged) TipR purified in 2 steps from an *E. coli* overexpression strain (see Experimental procedures). We observed a retardation of the labelled probe by TipR in a concentration-dependent fashion, even in the presence of nonspecific calf thymus competitor DNA (Fig 3A). EMSA experiments with *E. coli* extracts containing *WT* TipR also revealed retardation of Cy5-labelled P$_{acrA}$ compared to *E. coli* extracts without TipR. Next, we probed *E. coli* extracts each containing a different TipR mutant described above (E119V, S53R, or L204Q) for binding of P$_{acrA}$ by EMSA. While the L204Q variant retards P$_{acrA}$ akin to *WT* TipR, the E119V or S53R mutations abolish P$_{acrA}$ binding (Fig 3A). Immunoblots conducted with *C. crescentus* cell extracts revealed the E119V and S53R variants to accumulate to the same steady-state level as *WT* TipR upon NAL induction (S8A Fig), indicating that the E119V and S53R mutations promote DNA binding. By contrast, the abundance of TipR-L204Q is severely reduced, even in the presence of NAL (S8A Fig), suggesting that this mutation interferes with protein stability rather than with DNA-binding activity as shown by antibiotic chase experiments, revealing a reduced half-life of TipR-L204Q (Fig 3B).

Having identfied determinants in TipR that promote DNA binding, we next determined the effects of the isolated promoter mutations. To this end, P$_{acrA}$ promoter mutants with alterations in the IR were labelled with Cy3 and incubated with purified WT TipR in EMSAs (Fig 3C). While P$_{acrA}$-RIR is no longer retarded by TipR in EMSAs, we only observed a partial loss of binding by TipR to P$_{acrA}$-MIR. As expected, TipR still retarded the P$_{acrA}$-IR$^{up}$ probe that carries a mutation outside the IR. Thus, these EMSAs validate the requirement of the IR as the likely TipR docking site in P$_{acrA}$. With these results, we investigated further how P$_{acrA}$-IR$^{up}$ can cause overexpression of the pump without crippling DNA binding of TipR. EMSAs with a new Cy5-P$_{acrA}$ probe combining the P$_{acrA}$-RIR mutant (that abolish binding of TipR) with the P$_{acrA}$-IR$^{up}$ mutation, EMSA (Fig 3D) revealed that P$_{acrA}$-IR$^{up}$ increased the affinity of TipR for the P$_{acrA}$-RIR probe, suggesting that P$_{acrA}$-IR$^{up}$ mutation creates a secondary docking site upstream of the IR.

Indeed, the consensus target sequence derived for TipR (S9 Fig) resembles the motif created by the P$_{acrA}$-IR$^{up}$ mutation. In search for additional binding sites of TipR, we conducted ChIP-Seq experiments (S3 Data) and found that in vivo TipR binds 4 sites on the chromosome: the promoter of *acrAB-nodT* and *tipR*, the promoter of the *djlA* gene predicted to encode a DnaJ-like co-chaperone, as well as a site upstream of the DNA methyltransferase gene *ccrM* and another site in between the *qor* and *rho* genes oriented towards each other. The RNA-Seq experiments described above confirmed the NAL induction of 3 genes neighbouring two of the TipR binding sites (*acrAB-nodT*, *tipR*, and *djlA*), and ChIP-Seq experiments conducted with NAL-treated cells revealed that the binding of TipR to all 4 chromosomal sites is impaired in vivo (Figs 1F and S10). However, the binding of TipR to P$_{acrA}$ was approximately 250% more efficient than to the other sites, suggesting that the IR in P$_{acrA}$ represents the preferred recognition sequence of TipR. Nonetheless, we scanned the other target sites for sequences comparable to the IR and detected one upstream of *djlA* and of *ccrM*, delivering the consensus sequence 5′-WTGaGWMtGAWC-3′ by MEME analysis (S9 Fig). Using this consensus, we analysed the intergenic sequence between the *qor* and *rho* genes and found a single

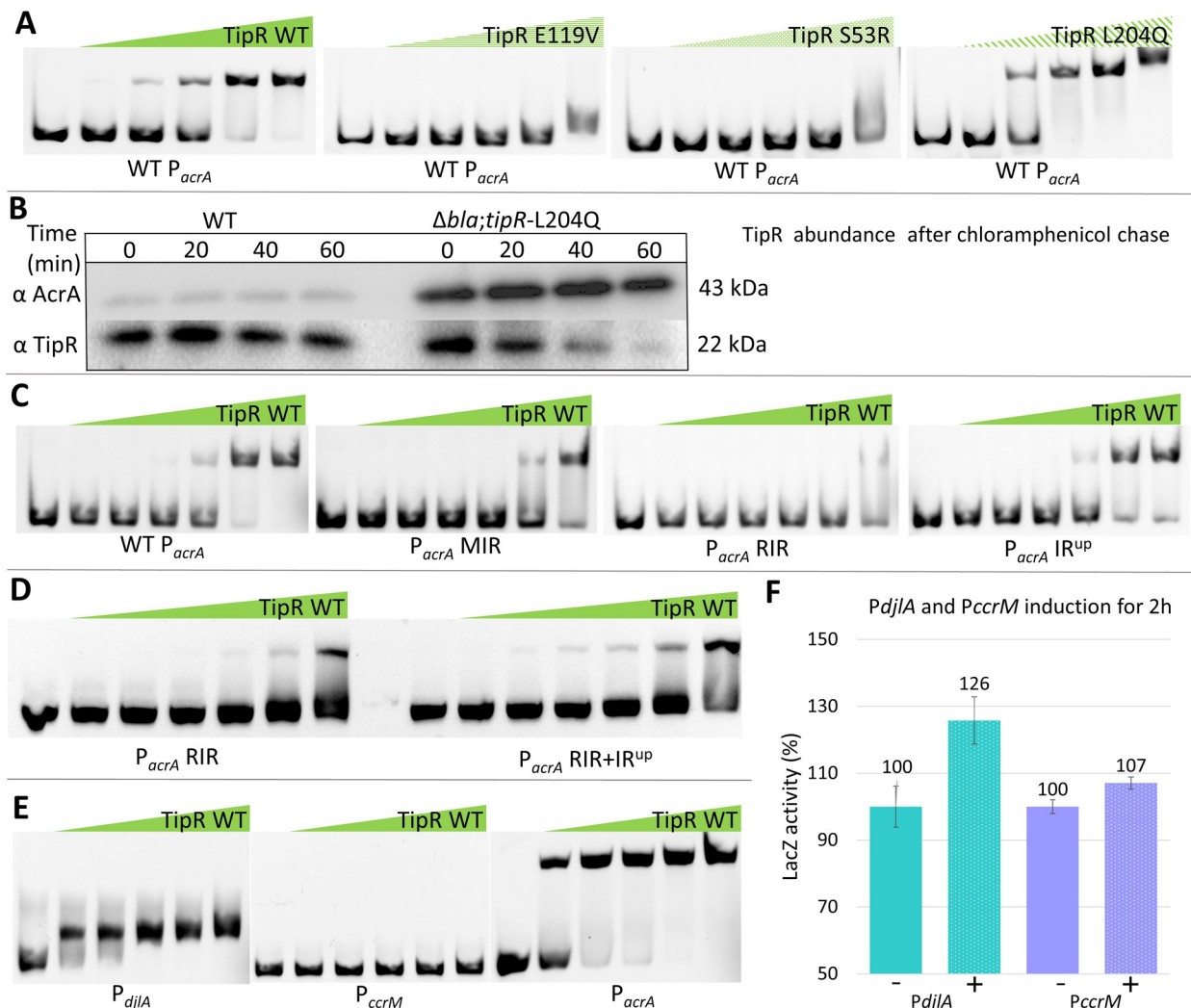

**Fig 3. Determinants in TipR and P$_{acrA}$ required for repression.** (**A**) EMSA with 200 ng of a Cy5-labelled *acrA* promoter (P$_{acrA}$) probe in the presences of increasing concentration of cell extract from *E. coli* overexpressing different mutated TipR protein from 0.031 μM to 16 μM. (**B**) Immunoblot showing the result of a 1-hour chase experiment using chloramphenicol (5 μg/mL) on exponential NA1000 (*WT*) and Δ*bla*;*tipR*-L204Q cells in PYE. (**C**) EMSA with 200 ng of different Cy5-labelled P$_{acrA}$ mutants analysed for retardation by gradual concentration of purified TipR protein from a concentration of 0.0156 μM to 16 μM. (**D**) EMSA with 100 ng of Cy5-labelled *acrA* with gradual concentration of heparin-purified TipR protein from a concentration of 0.25 μM to 8 μM. (**E**) EMSA with 100 ng of Cy5-labelled *djlA*, *ccrM*, and *acrA* promoters (P$_{djlA}$, P$_{ccrM}$, and P$_{acrA}$) with gradual concentration of purified TipR protein at a concentration of 1.125 μM to 18 μM. (**F**) Activity of β-galactosidase of the P$_{djlA}$-*lacZ* or P$_{ccrM}$-*lacZ* in *WT C. crescentus*. The « + » indicates induction by Nal (10 μg/mL) for 2 hours. All levels are indicated in percentage of expression regarding the basal level of NA1000 (*WT*) without induction. The data from the analysis are deposited in S2 Data. EMSA, electrophoretic mobility assay; PYE, peptone-yeast extract; WT, wild-type.

sequence that could explain the presence of TipR at this location; however, the configuration of those genes facing each other makes a TipR-dependent transcriptional regulation of these 2 genes unlikely.

Next, we tested the binding ability of TipR to *djlA* and *ccrM* promoters in vitro by EMSA using purified recombinant TipR. As seen in Fig 3E, TipR was able to retard the *djlA* promoter, but not the promoter of *ccrM*. Moreover, LacZ-based promoter probe assays with P$_{djlA}$-*lacZ* and P$_{ccrM}$-*lacZ* reporters revealed an induction of P$_{djlA}$ in *WT* cells in response to NAL, but no significant induction of P$_{ccrM}$-*lacZ* (Fig 3F and S2 Data). Since TipR binds P$_{ccrM}$ weakly in vivo, likely because of the divergence of its IR from the preferred TipR target sequence, the

weak interaction might result in a TipR-promoter complex too fragile to be maintained during electrophoresis. Moreover, in the absence of a significant induction of transcription by NAL in our RNA-Seq experiment (Fig 1C), we speculate that TipR is a poor repressor of $P_{ccrM}$ in vivo, yet it clearly down-regulates $P_{djlA}$.

## Screening for chemical inducers of $P_{acrA}$

Having identified the IR of $P_{acrA}$ as the target of TipR's DNA-binding activity, we engineered a reporter for rapid screening of chemical libraries for compounds that induce $P_{acrA}$ (independent of requirement of AcrAB-NodT efflux function as in CEF[10] resistance screen above). To this end, we fused to the promoterless *nptII* kanamycin resistance gene to $P_{acrA}$, creating a convenient readout for molecules like NAL that can activate $P_{acrA}$ in vivo scored as kanamycin resistance. We transformed the resulting $P_{acrA}$-*nptII* promoter probe reporter plasmid (p$P_{acrA}$-*nptII*) into *WT* cells, embedded the resulting reporter cells in soft agar containing kanamycin (10 μg/mL, KAN[10]) overlaid on KAN[10] plates, and then spotted 4 μL drops of each compound (at 10 μM) of the Maybridge chemical library for inducers of $P_{acrA}$ on these seeded indicator plates. Small molecule inducers were identified by their ability to promote growth of cells around the disc owing to activation of $P_{acrA}$-*nptII* by the compound (s) (S11 Fig). This chemical screen along with tests of candidate compounds reported in the literature for other systems, unearthed 14 different inducers of $P_{acrA}$-*nptII* (Figs 4A and S11) that were subsequently grouped into 2 classes based upon p$P_{acrA}$-*lacZ* inducer strength as determined by LacZ activity measurements. One group of strong inducers (triggering $P_{acrA}$-*lacZ* activity exceeding 230% activity relative to *WT*) includes the quinolones NAL, flumequine (FLU), and sparfloxacin, quinolone precursors such as chloroquine and dyes such as rhodamine 6G (Rh6) and crystal violet (CV). The weak inducers include the dye malachite green (MG), DNA intercalants such as acridine orange (AO), ethidium bromide (EtBr), and SYBR safe, but also various other compounds such as glafenine and doxylamine. We also noticed that dopamine and aminolevulinic acid fail to induce $P_{acrA}$-*lacZ* after 3 hours; however, they allowed growth on kanamycin plates with $P_{acrA}$-*nptII* after overnight incubation (Figs 4A and S11 and S2 Data), suggesting that they act slowly, through another pathway, or that they counteract kanamycin in an alternative way.

Some of the identified molecules are known substrates of the enterobacterial AcrAB-TolC RND pump and would thus likely be exported to reduce the active concentration in cells. Hence, we wondered if induction of $P_{acrA}$ would be accentuated in cells lacking AcrAB-NodT (S12A Fig and S2 Data) that can no longer expel these molecules, resulting in their accumulation to higher intracellular levels. While the level of $P_{acrA}$ induction with compounds like NAL or EtBr was similar in *WT* versus Δ*acrAB-nodT* cells, AO clearly showed a 4-fold elevated level of induction compared to the 2-fold induction in *WT* cells. Thus, the intracellular concentration modulates the level of *acrAB-nodT* expression. To eliminate the possibility that the compounds cause a nonspecific global increase of gene expression or LacZ enzymatic activity, we tested $P_{bla}$-*lacZ* reporter expressing LacZ expressed from the constitutive promoter of the *C. crescentus* MBL gene *CCNA_02223* and found that these compounds did not alter LacZ activity from $P_{bla}$-*lacZ* (S12B Fig and S2 Data).

## Different inducers control TipR at the post-translational level

To determine if the new inducers act in the same way on TipR as NAL, we first explored if they affect TipR stability in vivo. To this end, we uncoupled TipR synthesis from its inducible transcription by expressing it from the IPTG-controllable $P_{lac}$ promoter (of *E. coli*) on plasmid pSRK-*tipR* in *tipR*::Tn cells. AcrAB-NodT is still inducible from $P_{acrA}$ in these cells, but TipR

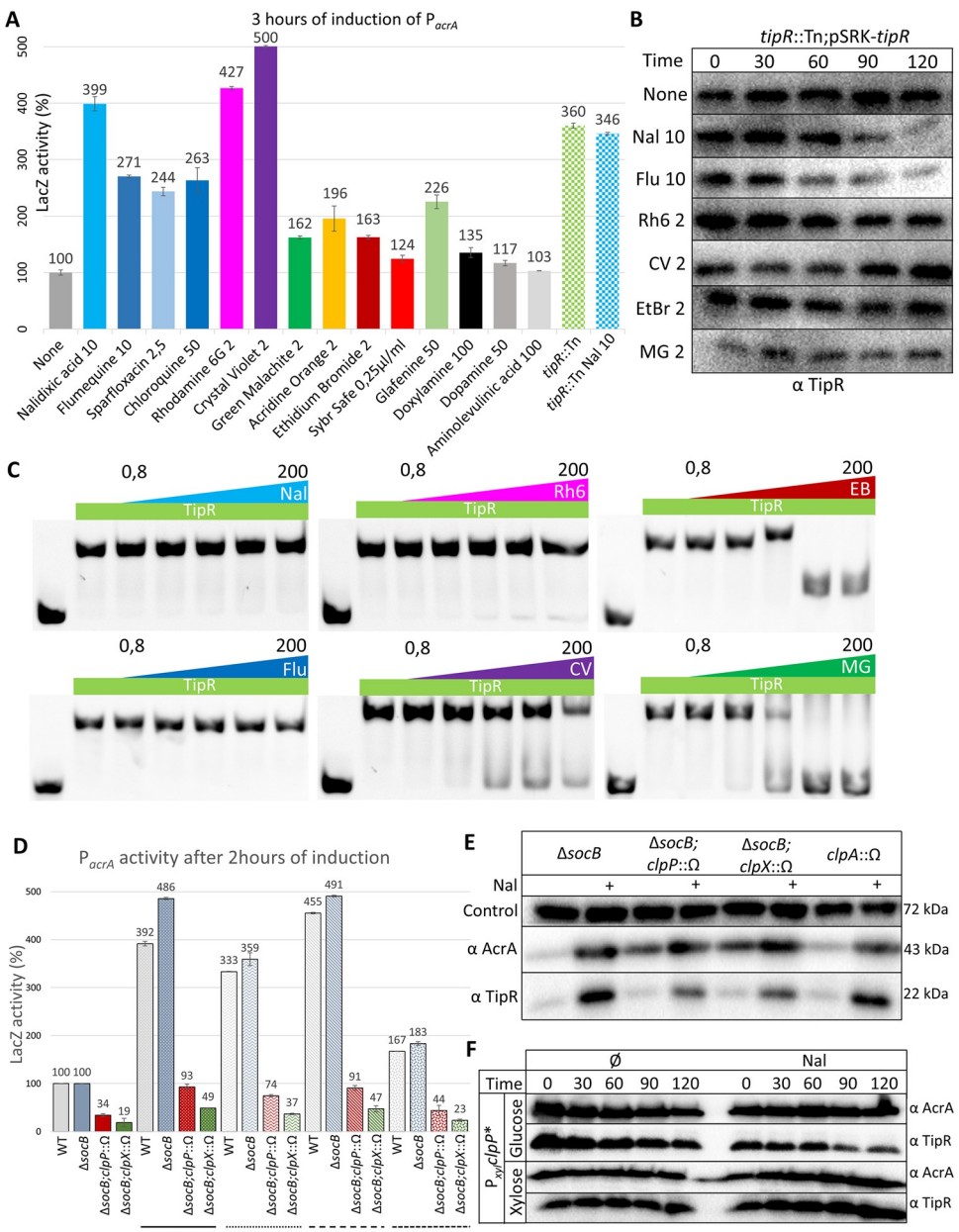

**Fig 4. Different chemicals relieve repression of P_acrA by TipR.** (**A**) Expression of β-galactosidase assayed from P_acrA-lacZ in NA1000 (*WT*) after 3 hours of induction by different compounds. All concentrations are in μg/mL. All levels are indicated as percentage of expression regarding the basal level of the control: NA1000 (*WT*) without induction. The data from the analysis are deposited in S2 Data. (**B**) Immunoblot revealing the abundance of TipR during a 2-hour chase experiment with chloramphenicol (5 μg/mL) added to exponential *tipR*::Tn;pSRK-*tipR* cells grown in PYE or and without selected inducers: Nal (10 μg/mL), Flu (10 μg/mL), Rh6 (2 μg/mL), CV (2 μg/mL), EtBr (2 μg/mL), and MG (2 μg/mL). (**C**) EMSA using 200 ng of Cy5-tagged DNA of the promoting region of *acrA* with 8 μM of heparin-purified TipR, supplemented with an increasing concentration of selected inducers. All quantities are in μg/mL. (**D**) β-galactosidase measurements in *C. crescentus WT* and different mutants containing P_acrA-lacZ before or after induction with Nal (10 μg/mL), Rh6 (Rho, 2 μg/mL Rho), CV (2 μg/mL), and MG (2 μg/mL) for 2 hours. All measurements are expressed as percentage of expression relative to basal activity of the control: *WT* or Δ*socB* cells before induction. The data from the analysis are deposited in S2 Data. (**E**) Immunoblot analysis of cell extracts from different protease mutants performed with polyclonal antibodies to TipR and to AcrA. All inductions (+) were performed after 2 hours of induction with Nal (10 μg/mL). Shown as loading control are blots with polyclonal antibodies to CCNA_00163. (**F**) Immunoblot following a 2-hour chase experiment with chloramphenicol (5 μg/mL) on exponential cells of *WT* with a *clpP** inserted at the P_xyl locus. Cells were grown in PYE, followed by the addition of

glucose or xylose, and, subsequently, the inducer was added: Nal (10 μg/mL), and, finally, protein synthesis was blocked with chloramphenicol. CV, crystal violet; EMSA, electrophoretic mobility assay; EtBr, ethidium bromide; Flu, flumequine; MG, malachite green; Nal, nalidixic acid; PYE, peptone-yeast extract; Rh6, rhodamine 6G; WT, wild-type.

synthesis is independent from the tester compounds such as NAL, Rh6, and MG (S13A and S13B Fig). We then compared the half-life of TipR in these cells after exposing them to the chemical inducers. To stop translation of TipR, we treated cells with high levels of the protein synthesis inhibitor chloramphenicol, a phenicol antibiotic that is (likely) not a substrate of the AcrAB-NodT efflux pump. In the presence of the quinolones NAL or FLU, TipR is rapidly turned over (Fig 4B), reminiscent of the instability observed for TipR-L204Q. Such an induced instability was not observed with the other $P_{acrA}$-activating compounds.

Next, we examined the effect of these compounds on the DNA binding ability of purified TipR by EMSA (Fig 4C). While at physiologically relevant concentrations, NAL or FLU did not perturb retardation of the Cy-labelled $P_{acrA}$, other inducers such as EtBr or MG interfered with binding of TipR to $P_{acrA}$. Rh6 and CV affect TipR binding in the same manner, but less efficiently in vitro compared to EtBr or MG. To assess the specificity of these compounds on TipR, rather than interfering with protein–DNA interactions in a general fashion, we conducted control EMSA using the histone-like IHF protein and its target DNA sequence *attR* (S14 Fig) as probe. Increasing concentration of CV did not prevent IHF binding even at a concentration sufficient to change the charge of the probe resulting in its upward migration in the gel during electrophoresis. Therefore, CV, Rh6, EtBr, and MG apparently interfere with TipR's ability to bind the IR in $P_{acrA}$, explaining the derepression of $P_{acrA}$ by these compounds in vivo.

To rule out the possibility that induction of AcrAB-NodT by NAL or FLU acts through the SOS (DNA damage) response by corrupting DNA gyrase (encoded by *gyrAB*) as in *E. coli*, we confirmed that inhibition of GyrA by the fluoroquinolone ciprofloxacin (CIP) or of GyrB by the aminocoumarin novobiocin (NOV) alone did not induce of $P_{acrA}$ (S15 Fig and S2 Data). Moreover, NAL and CIP did not synergise to enhance $P_{acrA}$ induction, yet NAL and NOV together resulted in elevated $P_{acrA}$-*lacZ* induction. Consistent with these findings, we next assayed $P_{acrA}$-*lacZ* activity in *WT* cells expressing an additional GyrA variant: GyrA from *Brucella melitensis* or a mutant variant of *C. crescentus* GyrA, GyrA*(F96D) that is NAL sensitive and causes DNA damage and the SOS response in the presence of NAL. *C. crescentus WT* cells expressing either of these GyrA variants from pMT335 still induce $P_{acrA}$ upon the addition of NAL (S15 Fig and S2 Data), even more efficiently than *WT* cells. In fact, the level of NAL induction in cells expressing the NAL-sensitive forms of GyrA attains the level of induction when NAL and NOV are added jointly to *WT* cells, indicating that Gyrase inhibition can enhance AcrAB-NodT induction, but only after repression of $P_{acrA}$ by TipR has been relieved by the addition of NAL.

Prompted by the observation that, at physiological concentrations, NAL and FLU reduce the half-life of TipR without interfering with TipR binding to $P_{acrA}$ in vitro in EMSAs (Fig 4B), we speculated that a transient destabilization by a dedicated protease could account for the induction of AcrAB-NodT. Using a candidate approach, we then tested for protease(s) conferring instability of TipR by immunoblotting using polyclonal antibodies to TipR and AcrA (S16 Fig). While no significant changes in AcrA or TipR steady-state levels was observed in *ftsH* and *lon* mutant cells compared to *WT* cells, loss of ClpP or ClpX led to an accumulation of AcrA and a reduction in TipR levels in the absence of NAL. Indeed, previous work showed that AcrA and AcrB are substrates of the ClpP degradation machinery [22]. Yet, AcrA and TipR are still inducible upon exposure of *clpP* and *clpX* mutant cells to NAL (Figs 4E and S13),

but promoter probe assays using the P*acrA*-*lacZ* reporter revealed a strong reduction of LacZ induction when *clpP* and *clpX* mutant are exposed to NAL, Rh6, CV, or MG compared to *WT* cells (Fig 4D and S2 Data). Taken together, our findings suggest that the ClpXP protease promotes high level of induction of *acrAB-nodT* and *tipR*, whereas inactivation of the *clpA* chaperone gene in *WT* cells did not cause a major AcrA or TipR abundance before after induction with NAL (Fig 4E).

To test if ClpP controls the half-life of AcrA and TipR, we conducted antibiotic chase experiments in cells expressing a dominant negative version of ClpP (ClpP*) in which the catalytic serine is mutated to alanine (Figs 4F and S11A). This ClpP* variant is expressed from the xylose-inducible P*xyl* promoter at the *xylX* locus (*xylX*::P*xyl*-*clpP**). Immunoblotting using antibodies to TipR and AcrA revealed that TipR is turned over normally upon the addition NAL in the presence of glucose (to repress expression of ClpP* from P*xyl*). However, when cells are grown in PYE containing xylose to induce of the dominant negative ClpP* variant), TipR stability is increased. Lastly, to demonstrate the interaction of TipR and AcrA with ClpXP biochemically, we used a GFP Trap matrix to pull down a ClpX-YFP fusion protein from lysates prepared from *WT xylX:.Pxyl-ClpX-YFP* cells before or after induction with NAL. Immunoblotting revealed the presence of TipR and AcrA in the pulled down material (S17 Fig) regardless of the presence of NAL.

In summary, while some compounds dislodge TipR from the IR sequence, others like first-generation quinolones act to destabilize TipR. The ClpXP protease confers partial instability of TipR in the presence of NAL, and it also reduces TipR and AcrA steady-state levels in the absence of NAL. Lastly, we discovered that ClpXP is required for efficient transcriptional induction of *acrAB-nodT*.

## Co-induction and interaction of DjlA and AcrAB-NodT

Since DjlA, a putative DnaJ-like co-chaperone, is co-induced with AcrAB-NodT and TipR, we investigated whether DjlA interacts with TipR or AcrAB-NodT or affects their function. We first examined whether DjlA overexpression alters induction of P*acrA* or TipR and AcrA steady-state levels upon addition of NAL. Neither P*acrA*-*lacZ*-activity nor AcrA or TipR abundance was affected as a function of DjlA, regardless of whether NAL was present or not (Figs 5A and S18 and S2 Data). However, when we probed for AcrAB-NodT-mediated efflux activity, we found that ectopic expression of DjlA enhances resistance to several β-lactam antibiotics belonging to the cephalosporin and penicillin substrates of AcrAB-NodT (S19 Fig). To assess if DjlA also promotes AcrAB-NodT-dependent efflux of other substrates, we conducted survival (efficiency of plating (EOP)) assays on plates containing EtBr[6] (EtBr, 6 μg/mL) and found that cells ectopically expressing DjlA (from pMT335-*djlA*) exhibited an elevated EOP compared to *WT* cells with the empty vector (Fig 5B). The positive effect of DjlA was also observed by a reduction of Δ*djlA* cells, survival on EtBr[4] (EtBr, 4 μg/mL) plates. This effect was restored in the presence of a complementing plasmid (pMT335-*djlA*) but completely lost when AcrAB-NodT was inactivated (i.e., in Δ*acrAB-nodT* cells; Fig 5B and 5C). Together, these results support the view that the activity or assembly of the pump, and thus the ability to expel EtBr, is enhanced by DjlA. Interestingly, forcing overexpression of AcrAB-NodT in Δ*djlA* cells by introducing the *tipR*::Tn mutation or by the addition of NAL can mitigate the reduced EOP on EtBr[4] plates, likely because AcrAB-NodT levels are no longer limiting for expulsion of EtBr to enable growth.

Since co-chaperones like DjlA typically assist in protein folding and/or disaggregation using the conserved DnaK chaperone, we constructed 2 DjlA variants with mutations in the HPD domain (H187A and H187Q) that is necessary for proper DnaK ATPase activity. Those

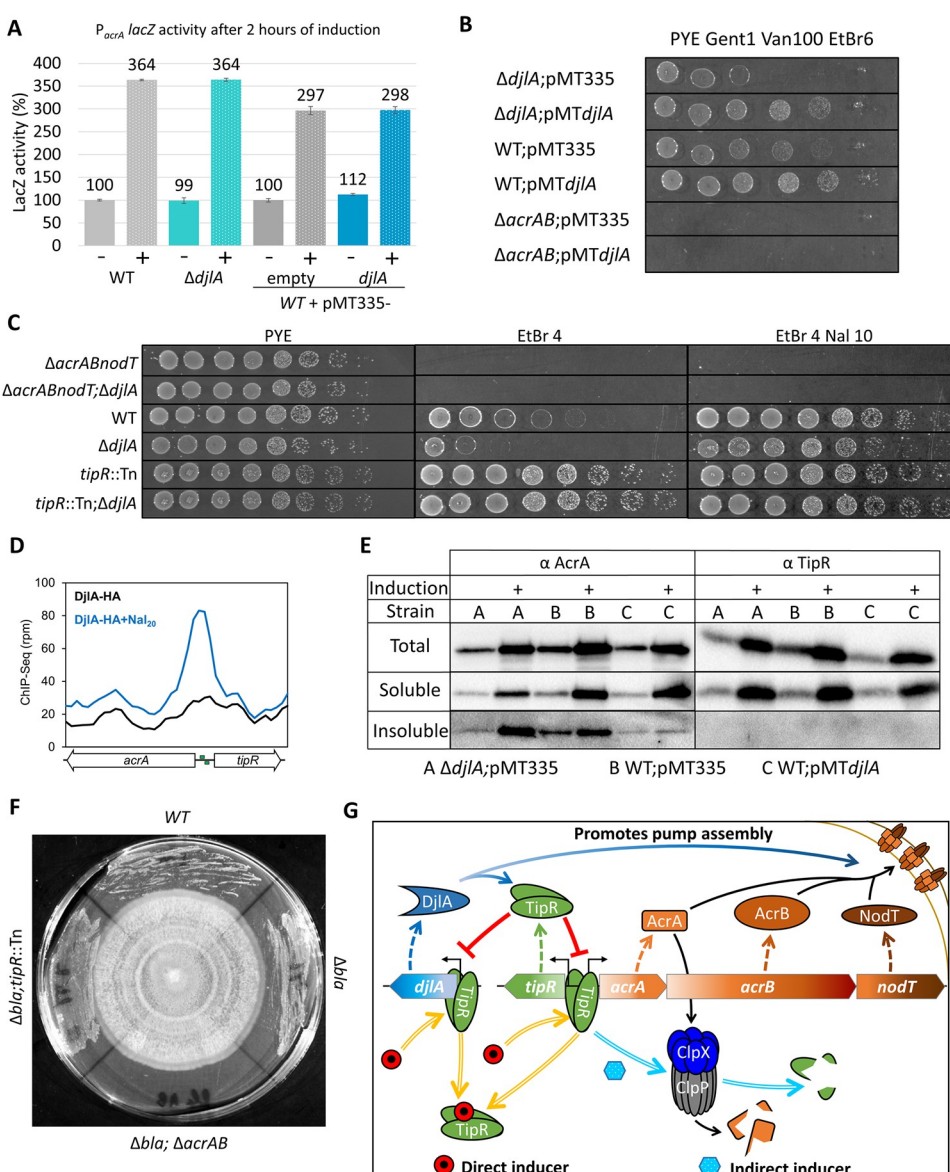

**Fig 5. Role of DjlA co-chaperone in the TipR-dependent control loop.** (**A**) Measurement of β-galactosidase expressed from P$_{acrA}$-*lacZ* in different *C. crescentus* Δ*djlA* mutants before or after a 2-hour induction with Nal [10 µg/mL (+) and/or vanillate 100 µM (V)]. All measurements are indicated as percentage relative to the basal activity of the *WT* control before induction. The data from the analysis are deposited in S2 Data. (**B**) EOP assay determined by 10-fold serial dilutions of Δ*djlA*::pNPTS and Δ*acrAB-nodT* mutants on plates containing EtBr (6 µg/mL) to probe for efflux pump activity. All strains contain pMT335 plasmid or a derivative (pMT335-*djlA*) expressing DjlA, grown on Van at 100 µM and Gent 1 µg/mL. (**C**) EOP assay by 10-fold serial dilutions of Δ*acrAB-nodT* and Δ*djlA*::pNPTS mutants on plates with EtBr (4 µg/mL) to probe for pump efficiency. Nal induction was performed at 10 µg/mL, Van at 100 µM, and Gent in µg/mL. (**D**) ChIP-Seq analysis of DjlA-HA expressed from the *xylX* locus in *WT* cells before and after NAL induction. DjlA-HA was pulled down using an anti-HA affinity matrix. The *acrA-tipR* promoter region was the only region showing a >2.9-fold increase in abundance compared to the control ChIP-seq performed with cells that were treated with NAL for 30 minutes or not treated. The data from the analysis are deposited in S3 Data. (**E**) Immunoblot analysis of aggregated cellular proteins in various strains: A = Δ*djlA*::pNPTS, B = *WT*, and C = *WT* pMT*djlA*. Inductions (+) were performed for 2 hours with Nal (10 µg/mL). The total fraction contains the cell lysate. The insoluble fraction corresponds to the material that remained in the pellet after several extractions with 1% Triton X-100. Strain C was grown in PYE with Gent (1 µg/mL) and Van (100 µM). (**F**) AcrAB-NodT enables growth of *C. crescentus* in the vicinity of an unknown fungus. *C. crescentus* strains grown in contact with an unknown fungus on a PYE plate. The fungus was pregrown on the plate for 2 weeks, after which the *C. crescentus* strains were streaked from the edge of the plate towards the centre until in contact with the fungus. The plate was then incubated at 30°C for another 72 hours. The fungus had been isolated in the laboratory as a contaminant on a PYE plate. AcrAB-NodT

encoded in the *C. crescentus* chromosome protects against (a) compound(s) produced by the fungus to enable growth of *C. crescentus* in the vicinity of the fungus. (**G**) Proposed model of the *acrAB-nodT* operon and *djlA* gene regulation, induction and function. EOP, efficiency of plating; EtBr, ethidium bromide; Gent, gentamicin; Nal, nalidixic acid; PYE, peptone-yeast extract; Van, vanillate; WT, wild-type.

2 constructs were put under the control of the inducible P$_{van}$ promoter on the pMT335 plasmid and transformed in strains lacking the endogenous *djlA* gene. The complementation assay performed on EtBr[6] agar plate demonstrates that the unmodified allele can complement the absence of the chromosomally encoded DjlA, while neither DjlA H187A or H187Q restore the phenotype (S20 Fig). Hence, the DnaK proper function is fully required for DjlA co-chaperone activity.

We reasoned that co-induction of DjlA and AcrAB-NodT by NAL may serve to manage a stress condition (for the envelope) imposed by the massive synthesis and assembly of the AcrAB-NodT structure, perhaps facilitating disaggregation of AcrA before secretion from the cytoplasm where DjlA is located. In support of this idea, pull-down experiments (S21A Fig) using a DjlA variant with a C-terminal HA-tag (DjlA-HA) revealed that DjlA directly or indirectly associates with AcrA and TipR. We conducted a similar experiment using the DjlA-H187A-HA variant in which a key residue affect the DnaK chaperone cycle is inactivated and found that the interaction of this DjlA-HA variant with AcrA and TipR still occurs, suggesting that the interaction of DjlA with AcrA and TipR does not depend on DnaK function (S21B Fig).

The interaction between DjlA-HA and TipR was further confirmed in ChIP-Seq (chromatin immunoprecipitation) experiments (Figs 5D and S22) revealing DjlA-HA can associate with TipR at the P$_{acrA}$ chromosomal site in vivo following exposure to NAL. In these experiments, crosslinked chromatin was prepared from DjlA-HA expressing cells after NAL treatment, followed by immunoprecipitation of DjlA-HA with bound chromatin using an anti-HA affinity matrix. Deep sequence analysis of the precipitated chromatin revealed a near 3-fold increase in P$_{acrA}$ abundance when DjlA-HA was immunoprecipitated from NAL-treated cells versus untreated cells. No association with P$_{acrA}$ was observed in control ChIP-Seq experiments with other HA-tagged DnaJ-like proteins. Moreover, the specific occupancy of DjlA-HA at P$_{acrA}$ required the presence of TipR (S22 Fig), as revealed by ChIP-seq experiments conducted in which *tipR*::Tn cells expressing DjlA-HA from the *xylX* locus had been treated with NAL. This result indicates that DjlA-HA specifically occupies P$_{acrA}$, that this association requires TipR, and that it is enhanced by NAL.

Finally, knowing that DjlA can interact with TipR and AcrA, we investigated whether DjlA also affects aggregation of TipR or AcrA. To test this, we first induced *WT* cells, Δ*djlA* cells, and DjlA-overexpressing cells (*WT*+ pMT335-*djlA*) with NAL and then separated extracts into soluble fractions and those that remain insoluble in Triton X-100 by centrifugation (Figs 5E and S23). Immunoblotting revealed approximately 40% less aggregated AcrA in the insoluble fraction from *WT* cells overexpressing DjlA after NAL[10] induction, while conversely, there was a 40% increase of AcrA aggregates induced by NAL[10] in Δ*djlA* cells. By contrast, no TipR was detectable in the insoluble fraction. To determine if DjlA also acts on other (envelope) proteins, we conducted LC-MS/MS (liquid chromatography followed by tandem mass spectrometry) analyses of the DjlA-HA pull-downs described above (S24 Fig). Pull-downs of DjlA-HA expressing cells grown in the absence of NAL revealed 8 of the 14 most expressed TonB-dependent receptors (TBDRs) and porins destined for the OM as clients of DjlA. In the presence of NAL[10], there was a drastic reduction in the number of OM proteins (OMPs) with a concomitant increase in abundance of AcrA, AcrB, and NodT, suggesting that DjlA has higher affinity for efflux pump components than for OMPs, which is consistent with its role in enhancing efflux activity by preventing its aggregation.

## Discussion

Here, we elucidated an integrated tripartite homeostatic loop (Fig 5G) controlling the transient induction of the AcrAB-NodT efflux pump in *C. crescentus*, the TipR repressor, and the DjlA co-chaperone from promoters that are TipR targets. We show that AcrAB-NodT induction confers adaptive resistance to β-lactam antibiotics from its TipR-repressed promoter that can be triggered by different classes of chemicals: Some are antibiotics, while others are general noxious molecules. Our work reveals 2 mechanisms controlling TipR derepression, with one class of compounds interfering with DNA binding of TipR, while others act on TipR stability. Importantly, we uncovered corresponding mutations in TipR by forward genetics that recapitulate these 2 divergent induction mechanisms. Our deep mutational scanning using forward selection for β-lactam resistance reinforced these findings and clearly unveiled 3 major functional domains in the TipR primary structure (Fig 2A): the N-terminal DNA binding, the central domain, and the extreme C-terminus for (de)stabilization. Mutations in each of these domains can lead to derepression of $P_{acrA}$ with overexpression of AcrAB-NodT and cephalosporin resistance. Lastly, our 2-step antibiotic selection regime and Mut-Seq showed that a second (elevated) level of resistance can still be acquired in *tipR* mutant cells that overexpress AcrAB-NodT by missense mutations in AcrB, augmenting cephalothin resistance to 3-orders of magnitude compared to regular AcrAB-NodT overexpression strains.

Although inducible overexpression of AcrAB-NodT is critical for survival against certain drugs or toxic molecules (also against natural molecules; Fig 5F), it also comes at a cost for envelope integrity and (cell shape) homeostasis, as well as OM barrier function. We discovered that long-term overexpression of the AcrAB-NodT envelope-spanning system results in strong OM perturbations that can sensitize cells towards antibiotics such as the CW-targeting antibiotic vancomycin that normally cannot traverse the OM. This barrier function is apparently compromised by excessive and/or long-term efflux pump (activity), perhaps owing to the OM detachment from other envelope layers membrane blebs and periplasmic vesicles that occur as revealed by cryoET. Such transient alterations from AcrAB overexpression could cause a discontinuous OM or openings through which drugs like vancomycin can subsequently enter. Thus, to minimize envelope integrity defects caused by long-term expression of AcrAB-NodT, cells must ensure that its expression is transient and that the repressed ground state can reestablish as quickly as possible after induction. Our findings in *C. crescentus* imply that clinical isolates whose MDR phenotype arises from massive overexpression of efflux pumps may also experience OM integrity defects, at least in the presence of antibiotics that induce the efflux pumps. In this case, such MDR isolates will be sensitized towards antibiotics like vancomycin when administered with inducers of efflux pumps.

The genetic linkage of the cytoplasmic co-chaperone DjlA with AcrAB-NodT and TipR is further reinforced by the biochemical interactions. Yet, we also found that DjlA also binds several TBDRs and other OMPs (see S24 Fig). Yet, upon AcrAB-NodT induction by NAL, DjlA is drawn to the preferred AcrAB-NodT substrate, in favour over other OMPs. Since DnaJ-type co-chaperones can sometimes (partially) substitute for one another [23,24], it is possible that AcrAB-NodT secretion or assembly depends on other co-chaperones. It is conceivable that such possible dependence of AcrAB-NodT on such co-chaperones is magnified in the absence of DjlA. Interestingly, DjlA has been implicated in promoting *trans*-envelope assembly and the envelope stress response in *Legionella pneumophila* and in *E. coli*, respectively [25,26]. Both these DjlA orthologs feature an N-terminal transmembrane domain, whereas no such segment is recognizable in *C. crescentus* DjlA [27]. Nonetheless, *C. crescentus* DjlA is shorter than typical γ-proteobacterial DjlA, and it is possible that it relies on another membrane association mechanism to capture and deliver its preferred substrates. The substrate preference

switch of *C. crescentus* DjlA towards AcrAB-NodT after NAL induction may accentuate the OM problems arising from extended efflux pump overexpression and assembly, perhaps leading to a reduction of insertion of OMPs that tether the OM to the CW and resulting in membrane blebs. In principle, we cannot rule out that blebbing arises from excessive translocation of phospholipids from the IM to the OM upon AcrAB-NodT overproduction, creating an overload that could lead to the OM invaginations or export of another substrate from the IM that affects OM tethering.

Our finding that the TipR is also induced by NAL, along with AcrAB-NodT and DjlA, is consistent with the view that an underlying tripartite homeostatic control loop sets the stage for reestablishing repression, for example, as soon as the inducing compound is expelled from the cytoplasm (Fig 5G). While repression by TipR will then terminate transcription of *acrAB-nodT*, the cell must also reduce the levels of the induced AcrAB-NodT protein by proteolysis rather than by dilution due to cellular division, which is substantially slower than proteolysis. Indeed, AcrA is substrate of the ClpXP protease [22], and our findings provide further evidence for interactions between TipR, AcrA, and the degradasome. The ATP-dependent protease ClpP restrict the passage to its proteolytic chamber only small, unfolded peptides, through narrow pores. To degrade bigger molecules, the proteasome needs ClpX that recognize specific substrates, directly or using adaptors to unfold the targeted protein using ATP. This then allows the denatured polypeptide chain to be translocated to the ClpP inner chamber where it gets degraded. One explanation could be that quinolones recruit an adaptor to TipR, or the antibiotic molecules block proper folding of the repressor, rendering them recognisable by the ClpXP machinery.

In the absence of a known natural inducer, we sought and discovered 14 mostly synthetic molecules that trigger derepression of TipR. Among the inducers described in this study, 2 molecules possess a curious behaviour: dopamine and aminolevulinic acid that do not induce AcrAB-NodT expression in the short term. While aminolevulinic acid can be used as a natural heme precursor, dopamine is a catecholamine characterised by an iron-binding domain. Both molecules are highly unstable and change conformation after time, in particular aminolevulinic acid that spontaneously dimerises into porphobilinogen, pseudoporphobilinogen or 2,5-dicarboxyethylpyrazine forming non-heme precursors [28]. As those molecules share structural similarities with efflux pump substrates, we speculate that aminolevulinic acid is not an inducer in this form, but its degradation or dimer products might trigger expression of the efflux pump, perhaps through an iron stress response. In support of this idea, our RNA-Seq analysis of NAL-treated cells shows an increase of expression in the transcript encoding the 5-aminolevulinic acid synthase, hinting at a purpose of *acrAB-nodT* expression control and inducers, with aminolevulinic acid derivatives activating the AcrAB-NodT expression to export an unusable product from the cell.

AcrAB-NodT likely exports natural products and metabolites upon induction with AcrB acting as key linchpin [29], a finding also implied from or CEF[40]-resistant *acrB* alleles. Chemical warfare between microbes typically involves a myriad of toxic molecules, sometimes targeting different compartments, and, thus, protection against many molecules at once can be signalled through a single inducer entering the cytoplasm. Indeed, an unidentified fungus isolated in the lab prevents growth of *C. crescentus* cells lacking AcrAB-NodT in proximity, but not cells expressing it (Fig 5F), thereby illustrating the importance of AcrAB-NodT in protecting bacteria towards natural toxic metabolites that likely also induce the system. However, we also noted that the contribution of AcrAB-NodT-mediated efflux is also detectable without NAL induction in assays of *WT* and for Δ*acrAB-nodT* cells using discs containing the cephalosporins cefepime or cefotaxime (S1B Fig). AcrAB-NodT also confers resistance to other antibiotics including erythromycin or other noxious compounds such as EtBr in the uninduced

state, indicating that AcrAB-NodT also expels compounds other than β-lactams. β-lactams do not induce AcrAB-NodT expression, a circumstance that is likely explained by the fact that these drugs are not known to enter the cytoplasm, unlike smaller molecules like EtBr or NAL. While this property explains why resistance to β-lactams in *C. crescentus* is an adaptive trait, it may also confer multiresistance to a wide range of toxic molecules with a minimal range of inducers of the efflux pump.

Collectively, our findings explain why AcrAB-NodT control on at least 2 levels (TipR and DjlA) is critical for transient efflux activation that protects bacterial cells in the wild from noxious molecules, but also from antibiotic treatment in the clinical setting.

## Materials and methods

### Growth conditions

All *C. crescentus* strains were cultivated in peptone-yeast extract (PYE) and incubated at 30˚C. ΦCr30-mediated generalized transductions were done as described [30]. *E. coli* strains were grown in lysogeny broth at 37˚C. All media were supplemented with the appropriate antibiotics or indicated inducer. The strain used for cloning is *E. coli* EC100D, while the strain used for protein expression is *E. coli* BL21(DE3). As both *WT* (NA1000) and Δ*bla C. crescentus* cells are naturally resistant to colistin (COL, 4 µg/mL) or aztreonam (AZT, 3 µg/mL), COL or AZT was used to counterselect *E. coli* Tn delivery strains for Tn (*himar1*) mutagenesis encoded on plasmid pHPV414 [31]. The *himar1* encodes resistance to kanamycin, and, therefore, the PYE plates were supplemented with kanamycin (20 µg/mL, KAN[20]) and CEF[10] or PIR[40]. Strains, plasmids, primers, and synthetic genes are detailed in S6 Data.

### β-galactosidase assays

All β-galactosidase assays were done using freshly electroporated strains harboring the pLac290-derived promoter probe plasmids to quantify the activity of the transcriptional fusions between the studied promoters and *lacZ*. The assays were done at room temperature. Between 50 and 100 µL of cells (OD$_{600}$nm of 0.3 to 0.7) were lysed in 30 µL of chloroform and vigorously mixed with Z buffer (60 mM Na$_2$HPO$_4$; 40 mM NaH$_2$PO$_4$; 10 mM KCl and 1 mM MgSO4 (pH 7)) to obtain a final volume of 800 µL. Followed by the addition of 200 µl of ONPG (Ortho-nitrophenyl-β-D-galactopyranoside, at 4 mg/mL in 0.1 M potassium phosphate (pH 7)) to begin the reaction. Assays were stopped using 500 µL of 1 M Na$_2$CO$_3$ when the solution turned light yellow. The OD$_{420}$nm of the supernatant was collected and used to calculate the Miller units as follows: U = (OD$_{420}$ * 1,000) / (OD$_{600}$ * t(min) * v(ml)). Error was calculated as standard deviation from at least 3 biological replicates (S2 Data).

### Kirby–Bauer disk diffusion susceptibility test

In a 14-cm diameter Petri dish, 50 mL of 1.5% agar media supplemented with indicated antibiotic and/or inducer were cast. After the polymerization, 12 mL of 0.375% agar media maintained at 50˚C were mixed with 400 µL of overnight bacterial culture prior to be spread evenly on top of the solid media. Antibiotics discs were from Bio-Rad or made in the lab using sterile disks. The pictures were taken after the plates were incubated overnight at appropriate temperate. Images were captured on a Bio-Rad illuminator and with the Image Lab 4.1 software.

### Efficiency of plating (EOP) assays

According to the OD$_{600}$, the same quantity cells from an overnight culture were placed in the first line of a 96-well plates. All the wells were filled with 180 µL of appropriate media except

the first line that was adjusted to 200 μL. The dilutions were done with 20 μL of the first line of wells transferred to the following line using a multipipette; this step was repeated until the 8 lanes were done. Finally, 5 μL of each dilution was dropped on a Petri dish with the solid media supplemented with the appropriate chemical(s).

## Microscopy

Around 2 μL of an exponential phase culture of *C. crescentus* grown in PYE supplemented with indicated chemicals were spotted on a thin layer of 1% agarose pad for immobilization. Contrast microscopy pictures were taken using a phase 100× objective with an oil interface (Zeiss, alpha plan achromatic 100×/1.46 oil phase 3) using an Axio Imager M2 microscope (Zeiss), with appropriate filter (Visitron Sys-tems GmbH) and a cooled CCD camera (Photo-metrics, CoolSNAP HQ2) controlled via the Metamorph software (Molecular Devices).

## Electrophoretic mobility shift assay (EMSA)

Cy5-labeled probes were generated by PCR with primers chemically modified with the fluores-cent dye. The probes were purified by agarose gel electrophoresis. The reaction took place in a buffer containing: 40 mM Tris-HCl (pH 7.6), 60 mM KCl, and 0.1% glycerol, 240 μg of calf DNA, 800 μg of BSA, and 200 ng of labeled probe. After the addition of the indicated quantity of purified protein, the samples were placed in the dark for 30 minutes at room temperature. After a prerun of 30 minutes, the samples were loaded and migrated by electrophoresis in a nondenaturing 4% Tris-Bore-EDTA acrylamide (19:1) gel. After migration, the samples were observed in a Bio-Rad illuminator (Chemidoc MP) using the preinstalled parameters for Cy5 imaging; pictures were collected through Image Lab 4.1 software (Bio-Rad).

## Chemical library screening

The Maybridge Chemical Library was delivered aliquoted into 96-well plates and dissolved in 50% DMSO. Petri dish of 22 cm containing PYE kanamycin 10 μg/mL were prepared using the Kirby–Bauer technique described above, with a NA1000 strain carrying the pLac290 plas-mid harboring a fusion between the P$_{acrA}$ and the *nptII* kanamycin resistance gene. Potential inducers were detected when a growth area occur at the position of a drop after 48 hours of incubation at 30˚C.

## Immunoblot analysis/chase

The appropriate strain was grown for 2 to 4 hours at 30˚C under constant agitation up to OD$_{600nm}$ of 0.4 to 0.6, then the inducer was added, and the cells were grown for 2 additional hours. For chase experiment, the strain was grown similarly up to OD$_{600nm}$ of 0.3 to 0.5, then chloramphenicol 5 μg/mL was supplemented with vigorous mixing, immediately followed by the addition of mentioned inducer. Protein samples from exponentially growing cells were sep-arated on a SDS–polyacrylamide (37.5:1) gel electrophoresis and blotted on 0.45 μm pore Poly-VinyliDenFluoride (PVDF) membranes (Immobilon-P from Sigma Aldrich). Membranes were blocked for 2 hours with 1× Tris-buffered saline (TBS) (50 mM Tris-HCl, 150 mM NaCl [pH 8]) that contain 0.1% Tween-20% and 8% powdered milk, followed by overnight incubation with the primary antibodies diluted in the same milk solution. The polyclonal antisera to AcrA (1:15,000 dilution), TipR (1:5,000 dilution), and CCNA_00164 (1:20,000 dilution) were used. The detection of primary antibodies was done using HRP-conjugated donkey anti-rabbit anti-body (Jackson ImmunoResearch) with Western Blotting Detection System (Immobilon from Milipore), and an imaging was performed in Bio-Rad illuminator (Chemidoc MP, Bio-Rad).

## Protein purification

TipR protein was expressed from pET21 in *E. coli* BL21(DE3)/pLysS. Cells were cultivated in LB at 37˚C and induced by 1 mM of IPTG for 4 hours, up to an OD$_{600nm}$ of 0.5. The bacteria were harvested at 8,000 RPM for 30 minutes at 4˚C. The pellet was resuspended in 25 mL of buffer (Tris-HCL 40 mM (pH 7.6), 50 mM KCl) before to be sonicated in a water–ice bath (15 cycles of 30 seconds ON, 30 seconds OFF). After centrifugation at 5,000$g$ for 20 minutes at 4˚C and filtration through 0.22 μm filters, the proteins contained in the supernatant were purified under native conditions successively using successively Qsepharose and Heparin columns (HiTrap systems from Cytia) according to manufacturer manual. Protein concentration was determined using the Bradford quantification method. Proteins were stored at −80˚C in 40 mM Tris-HCl (pH 7.6), 50 mM KCl, and 10% glycerol.

## Two-step resistance selection and Mut-seq

To isolate the Δ*bla;tipR*::*Tn* derivatives that grow on CEF[40] (Δ*bla;tipR*::*Tn acrAB-nodT**), Δ*bla; tipR*::*Tn* cells were plated on CEF[40], resistant mutant clones were pooled, and ΦCr30-lysates were prepared from this pool. The *tipR*::Tn allele was then retransduced into Δ*bla* cells, and transductants were selected on plates with KAN[20] and then tested for growth on plates with CEF[40]. The *acrAB-nodT* locus was sequenced in several such CEF[40]-resistant clones.

To isolate CEF[10]-resistant clones for Mut-Seq, the Δ*bla* strain was spread on PYE cephalothin 10 μg/mL and incubated at 30˚C for 48 hours. Around 10,000 colonies were harvested and pooled prior to DNA extraction using Ready-Lyse Lysozyme (Epicentre Lucigen) and DNAzol (Thermo Fisher). PCR were performed using the Q5 High-Fidelity DNA Polymerase according to manufacturer instructions with a maximum number of 15 cycles. The amplicons were purified with GeneJET Gel Extraction Kit (Thermo Fisher) and sent to sequencing at the iGE3 genomic platform in the CMU of the University of Geneva. The mixed amplicons were sent the Genomic platform iGE3 at the University of Geneva. Library preparation and sequencing were done using a HiSeq 2500 with 50-bp paired-end reads. Data analysis was done using Burrows–Wheeler Alignment Tool version 0.7.5a and Samtools version 1.2 prior to be blast against the *C. crescentus* NA1000 reference genome (NC_011916.1). SNPs with an abundance lower than 5% were considered background (PCR amplification errors) and excluded from the analysis. The analysis is presented in S4 Data.

## RNA extraction, deep- sequencing, and bioinformatics analysis

Overnight cultures in PYE of *C. crescentus* NA1000 (*WT*) were freshly restarted in 10 mL of PYE (starting OD$_{660nm}$ approximately 0.05) and incubated at 30˚C under agitation to reach an OD$_{600nm}$ of 0.5. NAL-treated cultures were supplemented with 20 μg/mL of antibiotic 30 minutes before being harvested. First, to provide immediate stabilization of RNA, each cell cultures (4 ml) are treated with 2 volumes of RNA Protect Bacteria reagent (Qiagen, Switzerland) according to the manufacturer's protocol. Then, cells were lysed in a Ready-Lyse lysozyme solution (Epicentre Technologies) according to manufacturer's instructions, and lysates were homogenized through QiaShredder columns (Qiagen, Switzerland). The RNeasy Mini Kit (Qiagen, Switzerland) was used for total RNA extraction according to the manufacturer's protocol, which included a first DNase treatment using the On-column DNase I digestion kit (Qiagen, Switzerland). Extracted total RNA was subjected to a second DNase treatment using Promega RQ1 at 1 unit/μg of RNA. Another total RNA cleanup was performed after the DNase treatment. The RNA concentration was measured using a Nanodrop 1000 spectrophotometer (Thermo Scientific, USA), and RNA quality was assessed using a 2100 Bioanalyzer

Instrument (Agilent Technologies, USA). Two independent biological replicates were analyzed per condition.

RNA-Seq library preparation and sequencing were performed at Fasteris SA (Geneva, Switzerland). Bacterial rRNAs were removed from each total RNA samples, and RNA-Seq libraries were prepared using the Ovation Complete Prokaryotic RNA-Seq library system kit according to the manufacturer's instructions. Single-end runs were performed on an Illumina NextSeq 500 instrument (50 cycles), yielding several million reads (stored as fastq files). Using the web-based analysis platform Galaxy (https://usegalaxy.org), the single-end sequenced reads quality was checked (FastQC, Galaxy Version 0.72), and reads were mapped (Bowtie2, Galaxy Version 2.3.4.3) to the *C. crescentus* NA1000 genome (NC_011916.1). The counts with the number of reads mapping to each gene feature were prepared using the htseq-count software, Galaxy Version 0.9.1. For each experimental series, the counts normalization and the statistical differential expression analysis was performed using the DESeq2 software, Galaxy Version 2.11.40.6. Sequence data (S1, S3, and S4 Data) have been deposited to the Gene Expression Omnibus (GEO) database (GSE225489 accession).

## Chromatin immunoprecipitation coupled to deep sequencing (ChIP-Seq) and data analysis

Overnight cultures in PYE of *C. crescentus* NA1000(WT) or Δ*bla;* P$_{xylX}$::P$_{xyl}$*djlA-HA* were freshly restarted in 80 mL of PYE (starting OD$_{600nm}$ of 0.05) and incubated at 30˚C under agitation with 0.3% xylose when necessary. Cultures were treated with 20 μg/mL of NAL 30 minutes before fixation. Cultures of exponentially growing cells (OD$_{600nm}$ of 0.5) were supplemented with 10 μM sodium phosphate buffer (pH 7.6) and then treated with formaldehyde (1% final concentration) at room temperature for 10 minutes to achieve crosslinking. Subsequently, the cultures were incubated for an additional 30 minutes on ice and washed 3 times in phosphate-buffered saline (PBS, pH 7.4). The resulting cell pellets were stored at −80˚C. After resuspension of the cells in TES buffer (10 mM Tris-HCl (pH 7.5), 1 mM EDTA, 100 mM NaCl) containing 10 mM of DTT, the cell resuspensions were incubated in the presence of Ready-Lyse lysozyme solution (Epicentre, Madison, WI) for 10 minutes at 37˚C, according to the manufacturer's instructions. Lysates were sonicated (Bioruptor Pico) at 4˚C using 15 bursts of 30 seconds to shear DNA fragments to an average length of 0.2 to 0.5 kbp and cleared by centrifugation at 14,000 rpm for 2 minutes at 4˚C. The volume of the lysates was then adjusted (relative to the protein concentration) to 1 mL using ChIP buffer (0.01% SDS, 1.1% Triton X-84 100, 1.2 mM EDTA, 16.7 mM Tris-HCl [pH 8.1], 167 mM NaCl) containing protease inhibitors (Roche) and precleared with 80 μl of Protein-A agarose (Roche, www.roche.com) and 100 μg BSA. Approximately 5% of each precleared lysate were reserved as total input samples (negative control samples). The precleared lysates were then incubated overnight at 4˚C with purified polyclonal rabbit anti-TipR antibodies (1:400 dilution) [9] or monoclonal rabbit anti-HA antibodies (1:250 dilution) (Clone 114-2C-7, Merck Millipore). The immunocomplexes were captured after incubation with Protein-A agarose beads (presaturated with BSA) during a 4-hour incubation at 4˚C and then washed subsequently with low salt washing buffer (0.1% SDS, 1% Triton X-100, 2 mM EDTA, 20 mM Tris-HCl (pH 8.1), 150 mM NaCl), with high salt washing buffer (0.1% SDS, 1% Triton X-100, 2 mM EDTA, 20 mM Tris-HCl (pH 8.1), 500 mM NaCl), with LiCl washing buffer (0.25 M LiCl, 1% NP-40, 1% deoxycholate, 1 mM EDTA, 10 mM Tris-HCl (pH 8.1)) and, finally, twice with TE buffer (10 mM Tris-HCl (pH 8.1), 1 mM EDTA). The immunocomplexes were eluted from the Protein-A agarose beads with 2 times 250 μL elution buffer (SDS 1%, 0.1 M NaHCO3, freshly prepared) and then, just like total input samples, incubated overnight with 300 mM NaCl at 65˚C

to reverse the crosslinks. The samples were then treated with 2 μg of Proteinase K for 2 hours at 45°C in 40 mM EDTA and 40 mM Tris-HCl (pH 6.5). DNA was extracted using phenol: chloroform:isoamyl alcohol (25:24:1), ethanol-precipitated using 20 μg of glycogen as a carrier and resuspended in 50 μL of DNAse/RNAse free water.

Immunoprecipitated chromatins were used to prepare sample libraries used for deep sequencing at Fasteris SA (Geneva, Switzerland). ChIP-Seq libraries were prepared using the DNA Sample Prep Kit (Illumina) following manufacturer instructions. Single-end run was performed on an Illumina Next-Generation DNA sequencing instruments (NextSeq High); 50 cycles were performed and yielded several million reads per sequenced samples. The single-end sequence reads stored in FastQ files were mapped against the genome of *C. crescentus* NA1000 (NC_011916.1) using Bowtie2 version 2.4.2.+galaxy0 available on the web-based analysis platform Galaxy (https://usegalaxy.org) to generate the standard genomic position format files (BAM). ChIP-Seq reads sequencing and alignment statistics are detailed in S3 Data. Then, BAM files were imported into SeqMonk version 1.47.2 (http://www.bioinformatics.babraham. ac.uk/projects/seqmonk/) to build ChIP-Seq-normalized sequence read profiles. Briefly, the genome was subdivided into 50 bp, and for every probe, we calculated the number of reads per probe as a function of the total number of reads (per million, using the Read Count Quantitation option). Using the web-based analysis platform Galaxy (https://usegalaxy.org), TipR ChIP-Seq peaks were called using MACS2 Version 2.1.1.20160309.6 (no broad regions option) relative to the total input DNA samples. The q-value (false discovery rate (FDR)) cutoff for called peaks was 0.05. Peaks were rank ordered according to their fold-enrichment values (S3 Data). Peaks with a fold-enrichment values >4 for TipR were retained for further analysis. Consensus sequences common to the 4 enriched TipR-associated loci were identified by scanning peak sequences (+ or − 75 bp relative to the coordinates of the peak summit) for conserved motifs using MEME (http://meme-suite.org/) [32]. Sequence data (S3 Data) have been deposited to the Gene Expression Omnibus (GEO) database (GSE225489 series).

### Purification of anti-TipR antibodies

We eluted high-affinity polyclonal antibodies to TipR from an immunoblot. To this end, 200 μg of purified TipR protein was blotted onto a PVDF membrane, and the blot was incubated overnight in 5 ml of anti-TipR serum at 4°C. The membrane was then washed 4 times with 5% fat-free milk and once with PBS. A band corresponding to the size of TipR was then cut out from the membrane and placed in a 15-ml falcon tube. The band was then incubated with 1 mL of buffer 1 [50 mM Glycine (adjusted to pH 2.8 with HCl), 500 mM NaCl] for 3 minutes prior to neutralization with 8 μL of Tris-HCl (pH 8.2). The eluate was harvested, and the membrane was incubated again in 1 mL of buffer 2 [50 mM Glycine (adjusted to pH 2.2 with HCl), 500 mM NaCl] for 3 minutes prior to neutralization with 16 μL of Tris-HCl (pH 8.2). The second eluate was combined with the first and used as purified antibodies at a 1:500 dilution for immunoblotting and 1:400 dilution for ChIP-Seq experiments.

### Plunge freezing of *Caulobacter crescentus*

*C. crescentus* cells were mixed with 10 nm Protein A conjugated colloidal gold particles (1:10 v/v, Cytodiagnostics), and 4 μl of the mixture was applied to a glow-discharged copper EM grid (R2/1 or R2/2, Quantifoil). The grid was automatically backside blotted for 4 to 6 seconds in a Mark IV Vitrobot (Thermo Fisher Scientific) by using a Teflon sheet on the front pad and plunge frozen in a liquid ethane–propane mixture (37%/63%) cooled by a liquid nitrogen bath. Frozen grids were stored in liquid nitrogen until loaded into the microscope.

## Cryo-electron tomography

*C. crescentus* cells were imaged by cryoET [33]. Images were recorded on Titan Krios 300 kV microscopes (Thermo Fisher Scientific) equipped with a Quantum LS imaging filter operated at a 20-eV slit width and K2 or K3 Summit direct electron detectors (Gatan). Tilt series were collected using a bidirectional tilt-scheme from −60 to +60˚ in 2˚ increments. Total dose was 130 to 150 e⁻/Å, and defocus was kept at −8 μm. Tilt series were acquired using SerialEM [34], drift corrected using alignframes, reconstructed, and segmented manually using the IMOD program suite [35]. To enhance contrast, tomograms were deconvolved with a Wiener-like filter [36].

## In vivo protein aggregation assay

Exactly, not around 40 mL of exponentially growing cells ($OD_{600}$ between 0.4 and 0.6) were rapidly cooled in ice bath and pelleted by centrifugation (6,000*g*, 10 minutes). All steps were performed at 4˚C, and buffers were all supplemented with protease inhibitor cocktail from Roche (cOmplete tablets, EASYpack). The pellets were washed once in buffer A (50 mM Tris/HCl (pH 8.0), 150 mM NaCl) and resuspended in 300 μL of buffer A supplemented with 100 U/mL Ready-Lyse Lysozyme (Epicentre Lucigen), 10 μg/mL of DNAse I (Roche), and 10 μg/mL of RNAse A (Invitrogen). Cells were lysed in a Bioruptor (Diagenode) (set to high, 15 cycles for 30 seconds at 4˚C). Lysates were centrifuged (5,000*g*, 10 minutes) 2 times to remove nonlysed cells. The protein concentration of the lysate was quantified by Bradford assay, and an aliquot was harvested as the Total fraction. The remaining samples were centrifuged (14,000*g*, 30 minutes) to pellet the insoluble protein fraction. The supernatant was kept as the Soluble fraction. The pellets were then resuspended in 300 μL buffer A supplemented with 1% (v/v) Triton X-100 with and incubated for 1 hour on ice with regular vortexing prior to the sonication in a Bioruptor (set to high, 1 cycle for 30 seconds) followed by a centrifugation (20,000*g*, 20 minutes) for washing. This procedure was repeated 3 times. The protein pellet was resuspended in 100 μL 1xSDS loading buffer prior to heating to 95˚C for 10 minutes (S5 Data).

## Immunoprecipitation

A volume of 50 mL of exponentially growing cells were harvested by centrifugation (20 minutes, 4,500 rpm at 4˚C). The pellets were washed in 50 mL 1xPBS and centrifuged for 15 minutes with 6,000 rpm at 4˚C and then washed once in 1 mL 1xPBS before centrifugation (for 5 minutes, 14,000 rpm, at 4˚C). The pellets were resuspended in 1 mL TES (10 mM Tris-HCl (pH 7.5); 1 mM EDTA; 100 mM NaCl) containing protease inhibitor (cOmplete tablets, EASYpack from Roche) at 2 tabs per 50 mL and 100 U of Ready-lyse/ml of culture (Ready-Lyse Lysozyme from Epicentre Lucigen) prior to 10 minutes of incubation at room temperature. Each sample were supplemented with 50 μL NP-40 10% (AppliChem); 1 μL EDTA 0.5 M (pH 8.0) (Sigma); 10 μL $MgCl_2$ 1 M; 10 μL DNAse I (Roche) 10 mg/mL; 10 μL RNAse A (Invitrogen) 10 mg/mL and incubated 20 minutes at room temperature with constant agitation. The nonlysed cells were removed by centrifugation (10 minutes, 8,000 rpm at room temperature). The supernatant was harvested and centrifuged for 20 minutes with 14,000 rpm at 4˚C. The remaining supernatant was kept as IP input.

For each sample, 25 μL of GFP-trap (GFP-Trap_A from Chromotek) or anti-HA affinity matrix beads (Roche) was rinsed 4 times with 1 mL TES buffer and centrifugation 2 minutes at 3,000 rpm. Then, 25 μL of beads were added to each IP input and left overnight at 4˚C with agitation. The beads were harvested by centrifugation (2 minutes at 3,000 rpm) and washed 4 times with wash buffer (10 mM Tris-HCl 7.5; 150 mM NaCl; 0.5 mM EDTA; 0.5% n-dodecyl-

β-D-maltoside (DDM)). The beads were then resuspended in 40 µL of 2× SDS sample buffer [125 mM Tris-HCl (pH 6.8); 4% SDS; 20% glycerol; 10% β-mercaptoethanol; 0.004% bromophenol blue] and incubated 10 minutes at 95˚C. Each sample was centrifuged for 2 minutes at 3,000 rpm, and the supernatant was collected (without pipetting the beads). Samples of 5 or 10 µL of each collected fraction was analyzed by SDS-PAGE and Coomassie or immunoblotting.

## Supporting information

**S1 Fig. Resistance of *tipR* mutants to β-lactams.** (**A**) Growth of *C. crescentus* mutant strains on plates containing NAL[10] (nalidixic acid, 10 µg/mL), CEF[10] (cephalothin, 10 µg/mL), and/or PIR[40] (piperacillin, 40 µg/mL) for 3 days. (**B**) Antibiograms of *C. crescentus* strains on PYE. Antibiotic discs, from top left to bottom right: Pivmecillinam 20 µg, Mecillinam 10 µg, Ampicillin 100 µg, Ceftazidime 40 µg, Meropenem 10 µg, Amoxicillin 4 µg, Cephalothin 30 µg, Cefepime 30 µg, Piperacillin 100 µg, Oxacillin 5 µg, Doripenem 10 µg, Cefotaxime 30 µg, Moxalactam 30 µg, Imipenem 10 µg, Aztreonam 30 µg, Cephalexin 40 µg, Cefsulodin 30 µg.
(PDF)

**S2 Fig. RNA-Seq analysis of the *tipR-acrAB-nodT* locus.** (**A**) Representation of the reads [represented as reads per million (RPM)] obtained from the RNA-Seq experiment covering the *tipR* and *acrAB-nodT* region in the NA1000 strain (*WT*). Induction was performed on exponentially grown cells in PYE for 30 minutes with nalidixic acid (Nal, 20µg/mL). Red curves represent the reads in forward orientation (FW); blue curves represent the reads in reverse orientation (RV). The data from the analysis are deposited in S1 Data. (**B**) Scheme showing the location of the AcrB mutations conferring high level of cephalothin resistance (CEF[40]) to *Δbla;tipR*::*Tn* cells.
(PDF)

**S3 Fig. Induction of AcrA and TipR by NAL.** (**A**) Immunoblot with polyclonal antibodies to AcrA and to TipR to probe extracts from NA1000 (*WT*) and *tipR* mutant (*tipR*::Tn) cells exponentially grown in PYE before and after 2 hours of induction of Nal (20 µg/mL). Loading control represents CCNA_00163 revealed with antibodies to CCNA_00163. (**B**) Immunoblot polyclonal antibodies to AcrA and to TipR to probe extracts of cells harvested during a time course experiment on NA1000 (*WT*) for 2 hours in the presence and absence of nalidixic acid (Nal, 10 µg/mL) in PYE.
(PDF)

**S4 Fig. Antibiotic sensitivity of cells overexpressing AcrAB-NodT.** Kirby–Bauer-based disc diffusion assays to establish antibiograms of *C. crescentus* strains on PYE. Antibiotic discs, from top left to bottom right: Piramycin 100 µg, Vancomycin 30 µg, Flumequine 30 µg, Sparfloxacin 5 µg, Norfloxacin 5 µg, Kanamycin 20 µg, Neomycin 30 µg, Colistin 50 µg, Ciprofloxacin 5 µg, Teicoplanin 30 µg, Bacitracin 130 µg, Trimethoprim 5 µg, Azithromycin 15 µg, Rifampicin 30 µg, Clindamycin 2 µg, Erythromycin 15 µg, Fosfomycin 50 µg. Note: Kanamycin and neomycin resistance is conferred by the *nptII* gene located the transposon inserted in the *tipR* gene. Red circles demarcate the growth boundary for the reference strain.
(PDF)

**S5 Fig. Envelope defects caused by long-term overexpression of AcrAB-NodT.** (**A**) Light microscopy (phase contrast) images of NA1000 (*WT*) and *tipR*::Tn cells during exponential growth in PYE in the presence or absence of Nalidixic acid (Nal, 10 µg/mL), 1-(1-Naphtylmethyl)-piperazine (NMP quantities are indicated. (**B**) Cryo-ET images *C. crescentus* Δbla

cells (left), Δ*bla*; *tipR*::Tn (center), and Δ*bla*; pSRK-*acrAB-nodT* (right). Scale bar, 250 nm. CP, cytoplasm; IM, inner membrane; OM, outer membrane.
(PDF)

**S6 Fig. Membrane deformation induced by AcrAB-NodT overexpression.** Representative cryo-tomographic slices (left) and 3D renderings (right) of *C. crescentus* Δ*bla*;*tipR*::Tn (**A**) and Δ*bla*;pSRK-*acrAB-nodT* (**B**) cells. Scale bar, 250 nm. CP, cytoplasm; IM (dark blue), inner membrane; OM (blue), outer membrane; S-Layer (cyan); vesicles (purple).
(PDF)

**S7 Fig. P*acrA* activity in various *tipR* mutants.** β-galactosidase activity expressed from P*acrA*-*lacZ* in various mutants. Induction (+) was for 2 hours with nalidixic acid (Nal, 10 μg/mL). All levels are indicated as percentage of expression regarding the basal level of the *WT* (NA1000) without induction. The data from the analysis are deposited in S2 Data. All strains carry additionally the pP*acrA*-*lacZ* promoter probe plasmid.
(PDF)

**S8 Fig. Abundance of TipR and AcrA in various *tipR* and P*acrA* point mutants.** Immunoblots probed with polyclonal antibodies to AcrA and to TipR in extracts of different P*acrA* (**B**, **C**, and **D**) or *tipR* (**A**) mutants. All inductions (+) were performed after 2 hours of treatment with 10 μg/mL of Nal on exponentially grown cells in PYE. Blots were also probed with antibodies to CCNA_00163 as loading control.
(PDF)

**S9 Fig. Weblogo analysis of the TipR target DNA sequence.** An inverted repeat (IR) sequences was detected at each of the 3 TipR-binding positions on the chromosome, plus a half-site at the fourth target site. Consensus sequence of the IRs based on the sequences detected at position 399827, 927547, and 2365267 on the chromosome. This consensus has been identified using MEME software (mean *P* value: $4.48 \times 10^{-6}$) and drawn by WebLogo (crooks).
(PDF)

**S10 Fig. Occupancy of TipR on its chromosomal targets as determined by ChIP-Seq.** Representation of the reads [in reads per million (RPM)] obtained from the ChIP-Seq analyses covering the TipR binding regions. Positions are indicated under the graphic. Induction was performed for 30 minutes with Nal (20 μg/mL) in PYE (blue line) compared with the not induced condition (black line). The yellow boxes indicate the positions of the putative TipR binding site. The data from the analysis are deposited in S3 Data.
(PDF)

**S11 Fig. Screen for chemical inducers of P*acrA* using the P*acrA*-*nptII* reporter.** Reporter assay of P*acrA* activity fused with the *nptII* gene (P*acrA*-*nptII* conferring kanamycin resistance) cloned on plasmid plac290 (pP*acrA*-*nptII*). To identify inducers, chemicals were spotted on *WT* cells carrying the pP*acrA*-*nptII* reporter plasmid embedded on soft agar on PYE plates both containing with kanamycin (10 μg/mL). Plates were incubated for 2 days at 30˚C.
(PDF)

**S12 Fig. Quantification of P*acrA*-*lacZ* activity by different chemical inducers.** (**A**) β-galactosidase activity using the P*acrA*-*lacZ* in the Δ*acrAB-nodT* cells (**A**) or P*bla*-*lacZ* in *WT* cells. (**B**) Inductions were performed for 3 hours. All levels are indicated in percentage of expression regarding the basal level of the uninduced state. The data from the analysis are deposited in S2

Data.
(PDF)

**S13 Fig. AcrA abundance upon the addition of chemical inducers of P_{acrA}.** (**A**) Immunoblots probed with polyclonal antibodies to AcrA and to TipR to detect AcrA and TipR in *tipR*::Tn cells complemented with the pSRK-*tipR* plasmid, induced with the indicated amount of IPTG (in mM) and Nalidixic acid (Nal, 10 μg/mL) for 2 hours. (**B**) Immunoblot using polyclonal antibodies to AcrA and to TipR to probe extracts of multiple mutant strains grown in PYE with IPTG 0.5 mM, with and without 2 hours induction of Nal (10 μg/mL), rhodamine 6G (Rh6, 2 μg/mL), and malachite green (MG, 2 μg/mL).
(PDF)

**S14 Fig. Crystal violet does not affect binding of IHF to its target.** EMSA with 4 μM of IHF protein and 200 ng of Cy5-labelled *attR* DNA as probe. All quantities of crystal violet (CV) indicated are in μg/mL.
(PDF)

**S15 Fig. Effect of chemical inducers and *gyrA* variants on P_{acrA}-*lacZ*.** β-galactosidase activity measurements from P_{acrA}-*lacZ* in NA1000 (*WT*) carrying the pMT335 or a derivative to express a *gyrA*F96N (GyrA*) or *gyrA* from *Brucella melitensis* (*Bm*) grown in PYE Van 50 μM. Antibiotics were used at the following concentrations: nalidixic acid (Nal, 10 μg/mL), ciprofloxacin (Cip, 2 μg/mL), rhodamine 6G (Rho, 2 μg/mL), novobiocin (Novo, 10 μg/mL). All levels are indicated in percentage of expression regarding the basal level of *WT* before induction. The data from the analysis can be found in S2 Data.
(PDF)

**S16 Fig. Abundance of AcrA and TipR in protease mutants.** Immunoblot analysis using polyclonal antibodies to AcrA and to TipR to determine the steady-state levels of AcrA and TipR in various protease mutants of *C. crescentus*, before and after induction with nalidixic acid (Nal, +, 10 μg/mL) for 2 hours in PYE. Loading control is anti-CCNA_00164. All samples were loaded on the same immunoblot.
(PDF)

**S17 Fig. Pull-down of ClpX-YFP.** Coomassie Blue–stained PAGE (12% gel) (left) and immunoblot (right) of a ClpX-YFP co-immunoprecipitation (GFP Trap Matrix) with polyclonal antibodies to TipR and to AcrA. Induction with (+) or without (−) Nal (10 μg/mL).
(PDF)

**S18 Fig. Induction of AcrA and TipR in cells lacking DjlA.** Immunoblot analysis of extracts from *WT* and co-chaperone mutants using polyclonal antibodies to TipR and to AcrA. All inductions (+) were performed for 2 hours with 10 μg/mL of nalidixic acid (Nal). Strains carrying the pMT335 are induced with vanillate 100 μM (Van). Immunoblots performed with antibodies to CCNA_00163 serve as loading controls.
(PDF)

**S19 Fig. Antibiogram of cells lacking or overexpressing DjlA.** Antibiograms of *C. crescentus* strains using antibiotic discs, from top left to bottom right, Mecillinam 10 μg, Ceftazidime 40 μg, Meropenem 10 μg, Amoxicillin 20 μg, Cephalothin 30 μg, Cefepime 30 μg, Piperacillin 100 μg, Oxacillin 5 μg, Doripenem 10 μg, Cefotaxime 30 μg, Moxalactam 30 μg, Imipenem 10 μg, Aztreonam 30 μg, Cephalexin 40 μg, Cefsulodin 30 μg. Nal induction performed at 10 μg/mL. All plates with strains carrying a pMT335 or pMT335-*djlA* are supplemented with

vanillate 100 μM.
(PDF)

**S20 Fig. EOP assays with DjlA expressed from plasmids in *WT* and mutant cells.** Efficiency of plating (EOP) assay determined by 10-fold serial dilutions of Δ*djlA* mutant on plates containing ethidium bromide (EtBr, 6 μg/mL) to probe for efflux pump activity. All strains contain pMT335 plasmid or a derivative (pMT335-*djlA*, pMT335-*djlAH187A*, and pMT335-*djlAH187Q*) expressing DjlA, grown on vanillate (Van) at 100 μM and gentamicin (Gent) at 1 μg/mL.
(PDF)

**S21 Fig. Pull-down of DjlA-HA.** (**A**) Coomassie staining after 12% PAGE (left) and immunoblotting (right) of a DjlA-HA co-immunoprecipitation eluate (from anti HA affinity matrix) probed with polyclonal antibodies to AcrA and to TipR. Induction was with Nal (+) at 10 μg/mL for 2 hours. (**B**) Immunoblots probed with polyclonal antibodies to TipR and to AcrA after pull-down of HA-tagged protein from extracts of cells expressing DjlA-HA, DjlA-H187A-HA, DjlAH187Q-HA, DnaJ1-HA, and DnaJ2-HA using the anti HA affinity matrix. Induction was with Nal (+) at 10 μg/mL for 2 hours.
(PDF)

**S22 Fig. ChIP-Seq analysis of the divergent promoter for *acrAB-nodT* and *tipR*.** (**A**) Representation of the reads (in reads per million (RPM)) obtained from the ChIP-Seq analyses covering the TipR binding regions. Positions are indicated under the graphic. Induction was performed for 30 minutes with Nalidixic acid (Nal) 20 μg/mL in PYE (blue line) compared with the not induced condition (black line). The data from the analysis are deposited in S23 Fig. (**B**) Immunoblotting of extracts from cells expressing DjlA-HA, DnaJ1 to 5-HA and ClpX-HA used in the ChIP-Seq experiment. The blots were probed with polyclonal antibodies to TipR and monoclonal antibodies to the HA tag.
(PDF)

**S23 Fig. Quantification of AcrA aggregation in various mutants.** Graphical representation of 4 replicates of aggregation of assay quantified with ImageJ from immunoblotting of AcrA in soluble versus insoluble cell lysates as determined by immunoblotting using polyclonal antibodies to AcrA. All inductions (+) were performed for 2 hours with 10 μg/mL of Nal. Strains carrying the pMT335 are induced with vanillate (Van, 100 μM). All samples are normalized regarding the quantification of uninduced WT set at 100%. The data from the analysis are deposited in S5 Data.
(PDF)

**S24 Fig. LC-MS/MS analysis of DjlA-HA pull-downs.** Graphical representation of the total spectrum count of various proteins detected by LC-MS/MS upon immunoprecipitation of DjlA-HA from soluble cell lysates using anti-HA affinity matrix. The lysates used were from *WT* expressing DjlA-HA plus (blue) and minus (grey) NAL, and a *WT* control without DjlA-HA (red). All the TonB-dependant receptors (TBDRs) are annotated with their respective CCNA_#. The data from the LC-MS/MS analysis are deposited in S5 Data.
(PDF)

**S1 Data. RNA-Seq analysis after induction with NAL.**
(XLSX)

**S2 Data. Quantification of promoter probe assays.**
(XLSX)

**S3 Data. ChIP-Seq analysis after induction with NAL.**
(XLSX)

**S4 Data. Mut-Seq analysis of *acrAB-nodT* selected by cephalothin resistance of Δ*bla cells*.**
(XLSX)

**S5 Data. Quantification of AcrA aggregation assay.**
(XLSX)

**S6 Data. List of strains, plasmids, primers, and synthetic genes.**
(DOCX)

**S1 Raw images. Immunoblots and EMSA gels uncropped.**
(PDF)

## Acknowledgments

We acknowledge instrument access at the imaging platform ScopeM at ETH Zürich. JC wishes to acknowledge Sabine Quindou, Jamy Gourmaud, and Frédéric Courant for nurturing an entire generation's curiosity.

## Author Contributions

**Conceptualization:** Jordan Costafrolaz, Gaël Panis, Bastien Casu, Silvia Ardissone, Martin Pilhofer, Patrick H. Viollier.

**Data curation:** Jordan Costafrolaz, Gaël Panis, Bastien Casu, Silvia Ardissone, Laurence Degeorges, Martin Pilhofer, Patrick H. Viollier.

**Formal analysis:** Jordan Costafrolaz, Gaël Panis, Bastien Casu, Silvia Ardissone, Laurence Degeorges, Martin Pilhofer, Patrick H. Viollier.

**Funding acquisition:** Martin Pilhofer, Patrick H. Viollier.

**Investigation:** Jordan Costafrolaz, Gaël Panis, Bastien Casu, Silvia Ardissone, Laurence Degeorges, Patrick H. Viollier.

**Methodology:** Jordan Costafrolaz, Gaël Panis, Bastien Casu, Silvia Ardissone, Laurence Degeorges, Patrick H. Viollier.

**Project administration:** Jordan Costafrolaz, Martin Pilhofer, Patrick H. Viollier.

**Resources:** Jordan Costafrolaz, Gaël Panis.

**Supervision:** Martin Pilhofer, Patrick H. Viollier.

**Writing – original draft:** Jordan Costafrolaz, Patrick H. Viollier.

**Writing – review & editing:** Jordan Costafrolaz, Gaël Panis, Bastien Casu, Silvia Ardissone, Martin Pilhofer, Patrick H. Viollier.

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
