## [Editor Report · Decision Letter 0]

14 Feb 2023

Dear Dr. Viollier, 

Thank you for submitting your manuscript entitled "Co-chaperone-mediated post-translational control of efflux pump induction underlies adaptive β-lactam resistance in Caulobacter crescentus." for consideration as a Research Article by PLOS Biology.

Your manuscript has now been evaluated by the PLOS Biology editorial staff, as well as by an academic editor with relevant expertise, and I am writing to let you know that we would like to send your submission out for external peer review.

Once your full submission is complete, your paper will undergo a series of checks in preparation for peer review. After your manuscript has passed the checks it will be sent out for review. To provide the metadata for your submission, please Login to Editorial Manager (https://www.editorialmanager.com/pbiology) within two working days, i.e. by Feb 16 2023 11:59PM.

Kind regards,

Paula

---

Senior Editor

PLOS Biology

---

## [Decision Letter · Decision Letter 1]

31 Mar 2023

Dear Dr. Viollier,

Thank you for your patience while your manuscript "Co-chaperone-mediated post-translational control of efflux pump induction underlies adaptive β-lactam resistance in Caulobacter crescentus." was peer-reviewed at PLOS Biology. It has now been evaluated by the PLOS Biology editors, an Academic Editor with relevant expertise, and by several independent reviewers. 

In light of the reviews, which you will find at the end of this email, we would like to invite you to revise the work to thoroughly address the reviewers' reports.

As you will see below, both reviewers find the work interesting and novel, but they also raise some concerns that will need to be solved before further consideration. In particular, we think it is important that you quantify the Western Blots and test the involvement of a DnaK/Hsp70 chaperone, as requested by reviewer #2. Reviewer #1 considers that the membrane blebbing phenomenon that accompanies activation is a distraction from the principal message of the paper, and we think that it could be moved to the supplementary information. 

We also suggest a change in the title to make it more compelling. We suggest: "Adaptive β-lactam resistance in Caulobacter crescentus is mediated by post-translational control of efflux pump induction" or "Adaptive β-lactam resistance in Caulobacter crescentus depends on post-translational control of efflux pump induction by co-chaperones".

Given the extent of revision needed, we cannot make a decision about publication until we have seen the revised manuscript and your response to the reviewers' comments. Your revised manuscript is likely to be sent for further evaluation by all or a subset of the reviewers.

**IMPORTANT - SUBMITTING YOUR REVISION**

*Re-submission Checklist*

*Published Peer Review*

*PLOS Data Policy*

*Blot and Gel Data Policy*

Sincerely,

Paula

---

Senior Editor

PLOS Biology

REVIEWS:

Reviewer #1: TetR-like regulators.

Reviewer #2: Bacterial chaperones.

Reviewer #1: This interesting work describes a curious phenomenon whereby nalidixic acid induces resistance to certain beta-lactams in Caulobacter. The activation involves the reversal of repression by a DNA binding transcription factor called TipR with the resulting expression of an efflux system. A particularly interesting detail is the further involvement of a chaperone protein. Overall I find the data to be convincing and the story sufficiently novel to merit serious consideration at PLoS Biology. There is significant room for improvement, however. As presented, the manuscript is not at all suitable for publication. 

The authors describe is a membrane blebbing phenomenon that accompanies activation. They concede at one point that they don't understand what this is all about and must speculate as to their content (lines 427-429). To me this is a distraction from the principal message of the paper and I would withdraw it.

There is a tendency throughout to explain experimental results using adjectives rather than proper quantitative methods. There are many such instances however particularly egregious one is "pull-downs of DjlA-HA expressing cells grown in the absence of NAL, revealed an impressive number…". "Impressive" does not add anything - please give me a number. 

I must add that the manuscript was a chore to read. There is a fair number of typos for example ("ptentially", "we also to obtain clones" etc…). A fair bit of the data is presented as a list of experiments lacking in context or hypothesis ("next, we…."). Finally, they employ an astoundingly excessive number of acronyms that leaves us with many passages that are virtually impossible to understand. NAL, NOV, IR, Rh6, WT, CV, EOP etc..etc… The result is unintelligible messes such as "In fact, the level of NAL-induction in cells expressing the NAL-sensitive forms of GyrA attains the level of induction when NAL and NOV are added jointly to WT cells, indicating that Gyrase inhibition can enhance AcrAB-NodT induction, but only after repression of PacrA by TipR has been relieved by the addition of NAL" and "Rh6 and CV affect TipR binding in the same manner, but less efficiently in vitro compared to EtBr or MG". A substantial amount of re-writing will be required before this manuscript can go to press.

Reviewer #2: Bacterial efflux pumps can confer adaptive resistance to a broad spectrum of antibiotics and are thus active players in the ongoing AMR burden. The manuscript by Costafrolaz and colleagues investigates in details the regulation and the impact of the AcrAB-NodT efflux pump of the Caulobacter crescentus model bacterium on multi-drugs resistance, highlighting a novel role for the DjlA/HSP40 cochaperone of HSP70 in vivo. The authors showed that induction of the AcrAB-NodT by nalidixic acid antibiotic confers protection against several β-lactams. In this case, they found that nalidixic acid induces the derepression of the Parc promoter of acrAB-nodT, which is under the control of the TipR protein, a TetR-like repressor. Control by TipR was then investigated in details, revealing key residues of TipR and important responsive regulatory elements within the promoters and the intergenic region of AcrAB-NodT and TipR. Such analysis revealed that TipR also controls expression of the DjlA cochaperone. In addition, they found that the stress protease ClpXP participates both in the instability of TipR and the transcriptional induction of acrAB-nodT in the presence of nalidixic acid. Further genetic screens led to the identification of several chemicals that activate acrAB-nodT either by interfering with tipR DNA-binding or by affecting its stability. Remarkably, the authors also found that overexpression of the AcrAB-NodT efflux pump led to membrane defects, revealing intriguing interplay between an increased resistance related to pump overexpression and its consequence on the integrity of the outer membrane and the observed increased sensitivity to other type of drugs that would normally be harmless. Finally, the authors found that the TipR-regulated DjlA cochaperone interacts with the AcrAB-NodT pump and is capable of enhancing the resistance capacity to several B-lactams, thus revealing its impact on drug efflux.

The manuscript is well written and presents very interesting and novel data of general interest. Yet, additional controls need to be performed in order to strengthen the proposed role of the cochaperone DjlA.

Major comments:

1- Data form Figure 6E suggest that DjlA overexpression increases the soluble fraction of AcrA and that ArcA is more soluble in the WT than in the djlA mutants. However, the current data from the western blots are not convincing, with differences hardly visible. The authors should quantify the bands obtained for several independent experiments. In addition, it seems that DjlA overexpression does not significantly increase the soluble fraction but prevents the accumulation of the insoluble form of ArcA. The robustness of these data is critical in order to demonstrate that DjlA is indeed contributing to the efficient assembly/activity in vivo.

2-The role of the cochaperone is never presented within the context of its HSP70/DnaK partner. Providing evidences that we are indeed looking at a DnaK cochaperone activity and not at a DnaK-independent function is thus essential. Since DnaK is essential, the authors should test the impact of the canonical His to Gln loop substitution in the J-domain of DjlA in their drop test assay and/or in the fractionation experiments. Such a substitution in J-domain cochaperone is known to abolish the interaction with DnaK without perturbing membrane assembly or substrate interaction. This is a simple but powerful test to bring in the whole DnaK machine in enhancing pump activity, and thus confirm that the DnaK cochaperone activity is indeed responsible for the phenotypes observed.

3-The proposed model (Figure 6G) for TipR and AcrAB-NodT interaction with DjlA does not take into account the fact that DjlA is a membrane anchor protein. Would a membrane localization of DjlA fit in this model? This should be discussed and the model adapted accordingly.

Minor comments:

1-It is known that there is some overlap between J-domain cochaperones, especially in the presence of the stress-responsive DnaJ. The fact that endogenous DnaJ is present could explain the week impact of a single djlA mutation on pump activity. This should be discussed.

2-The previously described function of DjlA in the enhancement of recombinant membrane protein production and in the proper function of the Dot/Icm secretion complex should be discussed in light of the data presented in this work.

3-The ChiP-Seq data from Figure 6D suggest that DjlA-HA binds PacrA via TipR. Do the authors have evidence suggesting that DjlA-HA does not bind PacrA in the absence of TipR? This should be discussed.

---

## [Editor Report · Decision Letter 2]

29 Sep 2023

Dear Dr. Viollier,

Thank you for your patience while we considered your revised manuscript "Multivalent chemical co-activation of AcrAB-NodT and the co-chaperone DjlA confers adaptive β-lactam resistance in Caulobacter" for publication as a Research Article at PLOS Biology. This revised version of your manuscript has been evaluated by the PLOS Biology editors and the Academic Editor.

Based on our Academic Editor's assessment of your revision, we are likely to accept this manuscript for publication, provided you satisfactorily address the following data and other policy-related requests.

1. DATA POLICY:

A) Supplementary files (e.g., excel). Please ensure that all data files are uploaded as 'Supporting Information' and are invariably referred to (in the manuscript, figure legends, and the Description field when uploading your files) using the following format verbatim: S1 Data, S2 Data, etc. Multiple panels of a single or even several figures can be included as multiple sheets in one excel file that is saved using exactly the following convention: S1_Data.xlsx (using an underscore).

B) Deposition in a publicly available repository. Please also provide the accession code or a reviewer link so that we may view your data before publication.

Regardless of the method selected, please ensure that you provide the individual numerical values that underlie the summary data displayed in the following figure panels as they are essential for readers to assess your analysis and to reproduce it: Figures 1CDF, 3F, 4AD, 5AD, and Supplementary Figures S2A, S7, S10, S12AB, S15, S22A, S23, S24.

**Please also ensure that figure legends in your manuscript include information on where the underlying data can be found, and ensure your supplemental data file/s has a legend.**

We require the original, uncropped and minimally adjusted images supporting all blot and gel results reported in an article's figures or Supporting Information files. We will require these files before a manuscript can be accepted so please prepare and upload them now. We require this for: Figures 1E, 3ABCDE, 4BDEF, 5E, and Supplementary Figures S3AB, S8ABCD, S13AB, S14, S16, S17, S18, S21AB, S22B.

Please carefully read our guidelines for how to prepare and upload this data: https://journals.plos.org/plosbiology/s/figures#loc-blot-and-gel-reporting-requirements

3. We recommend to change the title to make it more accessible: "Adaptive β-lactam resistance in Caulobacter crescentus depends on post-translational control of efflux pump induction by the DjlA co-chaperone".

4. The academic editor suggests you consider deleting the comic in Figure 1A & leaving it for a graphical abstract, and that there's something unusual about reference 2.

We expect to receive your revised manuscript within two weeks.

*Published Peer Review History*

*Press*

Sincerely,

Paula

---

Senior Editor,

pjaureguionieva@plos.org,

PLOS Biology

---

## [Editor Report · Decision Letter 3]

17 Oct 2023

Dear Dr. Viollier,

Thank you for your patience while we considered your revised manuscript "Multivalent chemical co-activation of AcrAB-NodT and the co-chaperone DjlA confers adaptive β-lactam resistance in Caulobacter" for publication as a Research Article at PLOS Biology. This revised version of your manuscript has been evaluated by the PLOS Biology editors and many of the policy and editorial issues highlighted in our previous letter have not been addressed.

Please address the following data and other policy-related requests.

1. DATA POLICY:

A) Supplementary files (e.g., excel). Please ensure that all data files are uploaded as 'Supporting Information' and are invariably referred to (in the manuscript, figure legends, and the Description field when uploading your files) using the following format verbatim: S1 Data, S2 Data, etc. Multiple panels of a single or even several figures can be included as multiple sheets in one excel file that is saved using exactly the following convention: S1_Data.xlsx (using an underscore).

B) Deposition in a publicly available repository. Please also provide the accession code or a reviewer link so that we may view your data before publication.

Regardless of the method selected, please ensure that you provide the individual numerical values that underlie the summary data displayed in the following figure panels as they are essential for readers to assess your analysis and to reproduce it: **Figures 1CDF, 3F, 4AD, 5AD, and Supplementary Figures S2A, S7, S10, S12AB, S15, S22A, S23, S24.**

**Please also ensure that all figure legends (including supplementary figures) in your manuscript include information on where the underlying data can be found, and ensure your supplemental data file/s has a legend.**

**2. BLOT AND GEL REPORTING REQUIREMENTS:**

**We require the original, uncropped and minimally adjusted images supporting all blot and gel results reported in an article's figures or Supporting Information files. We will require these files before a manuscript can be accepted so please prepare and upload them now. We require this for: Figures 1E, 3ABCDE, 4BDEF, 5E, and Supplementary Figures S3AB, S8ABCD, S13AB, S14, S16, S17, S18, S21AB, S22B.**

Please carefully read our guidelines for how to prepare and upload this data: https://journals.plos.org/plosbiology/s/figures#loc-blot-and-gel-reporting-requirements

**3. We recommend to change the title to make it more accessible: "Adaptive β-lactam resistance in Caulobacter crescentus depends on post-translational control of efflux pump induction by the DjlA co-chaperone".**

4. The academic editor suggests you consider deleting the comic in Figure 1A & leaving it for a graphical abstract, and that there's something unusual about reference 2.

We expect to receive your revised manuscript within two weeks.

*Published Peer Review History*

*Press*

Sincerely,

Paula

---

Senior Editor,

pjaureguionieva@plos.org,

PLOS Biology

---

## [Editor Report · Decision Letter 4]

19 Oct 2023

Dear Dr Viollier,

Thank you for the submission of your revised Research Article "Adaptive β-lactam resistance from an inducible efflux pump that is post-translationally regulated by the DjlA co-chaperone." for publication in PLOS Biology. On behalf of my colleagues and the Academic Editor, Lotte Søgaard-Andersen, I am pleased to say that we can in principle accept your manuscript for publication, provided you address any remaining formatting and reporting issues. These will be detailed in an email you should receive within 2-3 business days from our colleagues in the journal operations team; no action is required from you until then. Please note that we will not be able to formally accept your manuscript and schedule it for publication until you have completed any requested changes.

PRESS

Sincerely, 

Paula 

---

Senior Editor

PLOS Biology
